# Cell type-specific epigenetic regulatory circuitry of coronary artery disease loci

Dennis Hecker [1,2,10], Xiaoning Song [3,4,10], Nina Baumgarten [1,2,10], Anastasiia Diagel [3,4,10], Nikoletta Katsaouni[1,2], Ling Li [3,4], Shuangyue Li[3,4], Ranjan Kumar Maji [1], Fatemeh Behjati Ardakani[1], Lijiang Ma[5], Zhaolong Li[4,6], Aldo Moggio[3,4], Daniel Tews [7,8], Hendrik Sager[3,4], Lars Maegdefessel [4,6], Martin Wabitsch [7,8], Johan L. M. Björkegren[5,9], Heribert Schunkert [3,4], Zhifen Chen[3,4] ✉ & Marcel H. Schulz [1,2] ✉

Coronary artery disease is the leading cause of death worldwide. Recently, hundreds of genomic loci have been shown to increase risk for the disease, however, the molecular mechanisms underlying signals from risk loci remain largely unclear. Here, we integrate the latest statistics of coronary artery disease genetics from over one million individuals with epigenetic data from 45 cell types to identify genes and transcription factors whose regulation is affected by variants. Applying two statistical approaches, we identify 1580 candidate disease genes, including 23.5% non-coding RNA genes. Enrichment analysis and phenome-wide association studies link the candidate genes to disease-specific pathways and risk factors. We conduct a proof-of-concept biological validation for the non-coding RNA gene *IQCH-AS1* via knockout in a human preadipocyte strain. Our study not only pinpoints CAD candidate genes in a cell type-specific manner but also highlights the roles of an understudied ncRNA gene in CAD genetics.

The past 15 years have witnessed an explosion of genetic discoveries on common complex diseases, mainly by genome-wide association studies (GWAS). For coronary artery disease (CAD), more than 300 gene loci have been associated with genome-wide significance[1–3], deepening the understanding of disease pathology. Beyond genetic discoveries, a major challenge is to fill the gap between genetic variants and disease risk. To address this, functional studies are indispensable for unveiling related molecular mechanisms and guiding novel diagnostic and therapeutic developments. Importantly, GWAS signals are most often found in non-coding regions of the genome, which

indicates a major role for altered gene regulation in the etiology[4]. In this respect, downstream causal genes, and cell types of action need to be elucidated allowing us to pinpoint the biological mechanisms underlying CAD. To date, efforts have been made to prioritize causal genes in CAD loci by various independent or joint methods by assessing 1) biological plausibility, 2) rare coding variant(s) associated with CAD, 3) likely pathogenic variant(s) relevant to CAD in ClinVar[5], 4) evidence from cardiovascular (CV) drug(s), 5) causality by Mendelian Randomization studies, 6) a protein-altering variant in high linkage disequilibrium (LD) with the sentinel CAD variant, 7) expression

---

[1]Department of Medicine, Institute for Computational Genomic Medicine, Goethe University Frankfurt, Frankfurt, Germany. [2]Deutsches Zentrum für Herz- und Kreislaufforschung (DZHK), Rhein-Main, Germany. [3]Department of Cardiology, TUM University Hospital German Heart Center, TUM School of Medicine and Health, Technical University of Munich, Munich, Germany. [4]DZHK, partner site Munich Heart Alliance (MHA), Munich, Germany. [5]Department of Genetics & Genomic Sciences, Institute of Genomics and Multiscale Biology, Icahn School of Medicine at Mount Sinai, New York, USA. [6]Molecular Vascular Medicine, Klinikum rechts der Isar - Technical University Munich, Munich, Germany. [7]German Center for Child and Adolescent Health (DZKJ), Partner Site, Ulm, Germany. [8]Division of Pediatric Endocrinology and Diabetes, Department of Pediatrics and Adolescent Medicine, Ulm University Medical Center, Ulm, Germany. [9]Department of Medicine, Karolinska Institutet, Karolinska Universitetssjukhuset, Huddinge, Sweden. [10]These authors contributed equally: Dennis Hecker, Xiaoning Song, Nina Baumgarten, Anastasiia Diagel. ✉e-mail: zhifen.chen@tum.de; marcel.schulz@em.uni-frankfurt.de

quantitative trait loci (eQTLs) in a CAD-relevant tissue, and 8) CV-relevant phenotypes in knockout mouse models[1,2,6]. However, there are still GWAS loci which could not be linked to genes with these methods. Among these methods, eQTL analysis contributed the largest number of candidate genes[2,3,7], yet eQTLs might explain only a small fraction of GWAS heritability plausibly[8]. More diverse functional genomic readouts beyond transcriptomics are urgently needed to identify disease mechanisms. The epigenome regulates gene expression in cells and engages the response of genetic variation to environmental changes. While sporadic studies have explored tissue-level chromatin states or epigenetic marks at CAD loci[1,9,10], epigenetics of CAD genetics is rather under-investigated.

In the current study, we systematically investigate the cell type-specific epigenetic circuit of CAD genetics by integrating the summary genetic statistics from over one million individuals with the regulatory elements of 45 types of disease-relevant cell types. In brief, we leverage a method for finding variants that potentially affect transcription factor (TF) binding in combination with cell type-specific interactions between regulatory elements and genes, as well as a gene-based association test. We identified 1580 candidate genes at CAD loci, out of which 798 were not detected by previously reported methods. Beyond commonly explored protein-coding candidate genes, our analyses allowed extensive examination of the CAD-associated non-coding RNA (ncRNA) genes which were otherwise depreciated, often due to insufficient sequencing depth in transcriptome-based causal gene prioritization or challenges in the interpretation of their function. Furthermore, biological validation was conducted on the novel candidate CAD gene *IQCH-AS1*, a long non-coding RNA (lncRNA), to confirm the reliability of our analysis and improve our knowledge of CAD mechanisms.

## Results

### Cell type-specific annotation of CAD-associated SNVs

To explore epigenetic and cell type-specific effects of CAD genetics, we obtained the latest summary genetic statistics of CAD GWAS from over one million human individuals and bulk or single-cell epigenetic data from 45 disease-relevant cell types (Fig. 1a)(Supplementary Data 1, 2). The cell types were manually prioritized based on potential clinical relevance and the final selection determined by data availability. 47,635 CAD-associated single nucleotide variants (SNVs) (≤ 1% FDR)[1] and 24,799 SNVs in LD ($R^2 \geq 0.8$) resulted in a total of 72,432 CAD-SNVs included in our analysis (Supplementary Data 3). Based on Ensembl Variant Effect Predictor (VEP)[11], the majority (67.1%) of CAD-SNVs were intronic variants and less than 10% were variants likely with a stronger effect size, such as missense, UTR or splicing region SNVs (Fig. 1b). Beyond the general annotation using VEP, we further investigated the intersection of CAD-SNVs with epigenetic profiles of the 45 cell types, which covered 12 cell lineages including lymphoid, hematological, myeloid, erythroid, endothelial, myogenic, neuronal, adipogenic, fibroblast, hepatic, epithelial, and mesenchymal cells. On average 121,376 cis-regulatory elements (CREs) per cell type were identified and, in total, 1,243,265 CREs from the 45 cell types. 18.73% of the CAD-SNVs were located in CREs, out of which 14.49% in intragenic CREs, and 4.24% in intergenic CREs (Fig. 1c). Among the CAD-SNVs in CREs (N = 13,563), 39.80% were affecting predicted transcription factor (TF) binding (N = 5397) (TF-SNVs), meaning that they significantly increase or decrease how well a TF binding motif matches to the DNA sequence (Fig. 1d). From the perspective of CREs, we found that 0.71% (N = 8864) of them contained a CAD-SNV and for 47.89% of those CREs (N = 4233) the CAD-SNV was also a TF-SNV. CREs that are shared across cell types were more likely to contain a CAD-SNV or a TF-SNV than CREs unique to a cell type (two-sided Fisher's exact test, both p-values ≤ 0.0001; log2-oddsratio for CAD-SNVs 1.46; log2-oddsratio for TF-SNVs 1.68). Vascular and immune cells were the leading cell types with regard to the fraction of their CREs that contain a TF-SNVs, such as endothelial

cells (ECs), smooth muscle cells (SMC), fibroblasts, T cells, monocytes, and neutrophils (Fig. 1e), in line with their crucial roles in cardiovascular health. 957 TF-SNVs were located in CREs of neurons, resonating with the roles of neuroimmune interfaces in atherosclerosis[12].

### Epigenetic data reveals novel candidate CAD genes

Based on the epigenetic data of the 45 cell types (Fig. 1e), we used two complementary approaches to prioritize candidate genes for CAD, namely, the Gene-Based Association Test (GATES)[13] and the gene prioritization using SNEEP[14] (Methods, Supplementary Fig. 1a, Supplementary Data 4). While GATES represents a commonly used method that aggregates all SNVs in a gene body under consideration of their LD structure, SNEEP looks at epigenetic consequences by identifying TF-SNVs and linking them to genes based on predicted CRE-gene interactions. Those interactions between CREs and genes were created for each cell type separately[15]. We obtained 1387 genes with GATES and 503 genes with SNEEP resulting in a total of 1580 candidate CAD genes (Supplementary Data 5). We compared our candidate genes with reported CAD genes by other genetic studies and genes identified via GWAS-eQTL colocalization analysis[16] using genotyped transcriptome data of CAD-relevant tissue types from the Genotype-Tissue Expression (GTEx)[17] and Stockholm-Tartu Atherosclerosis Reverse Network Engineering Task (STARNET) projects[18]. Out of 1580 genes, 782 overlapped with reported CAD genes or with CAD eQTL genes, representing a replication rate of genes of 49.49% relative to other studies or methods (Fig. 2a, and Supplementary Fig. 1b, c). 798 genes were novel CAD candidate genes prioritized by our epigenetics-GWAS integration. Gene set enrichment analysis showed *lipid metabolism*, and *TGF beta, VEGFA*, and *interleukin 11 signaling* to be the top pathways for CAD risk mediated by the 1580 candidate genes (Fig. 2b, Supplementary Data 6), in line with known pathways for the disease. Our analysis also pointed to an ncRNA-related pathway, *miRNA targets in ECM and membrane receptors*, for CAD. Our 1580 candidate genes contained 1208 protein-coding and 372 ncRNA genes (Fig. 2c, Supplementary Fig. 1c). Our epigenetic integration identified the highest fraction of ncRNA genes compared to the other three sources, including the well-known cardiovascular disease (CVD) ncRNAs *CDKN2B-AS1*[19–21] and microRNA132 (MIR132)[22] (Fig. 2c, and Supplementary Fig. 1c, 2b).

We investigated how many of our candidate genes are significantly differentially expressed in CAD cases compared to controls. To do so, we used two datasets. First, RNA sequencing of 145 early (controls) and 57 advanced (cases) human carotid plaque lesions from patients undergoing carotid endarterectomy (Munich Vascular Biobank[23,24]). Second, RNA data from five tissues from CAD cases and controls (STARNET[18]), namely visceral abdominal fat, subcutaneous fat, atherosclerotic aortic root, liver, and skeletal muscle (≥ 500 cases and ≥ 100 controls for each tissue). Remarkably, a total of 1276 of our candidate genes (81%), including 165 non-coding RNA genes, were differentially expressed in at least one tissue (Fig. 2d). Many genes were differentially expressed across multiple tissues, with 16% of our candidates being differential in all tissues from both cohorts and 29% in all tissues from STARNET[18] (Supplementary Fig. 1d).

To illustrate the expression of candidate CAD genes across different cell types and tissues, we determined their expression breadth in the IHEC EpiATLAS[25], which measures in how many cell types a gene is expressed (Fig. 2e). As expected, while most of the protein-coding genes were expressed across many cell types, ncRNA genes showed mostly cell type-specific expression. The 503 genes prioritized by SNEEP had an average of 4.5 CREs with TF-SNVs across all cell types (Fig. 2f). The number of identified CAD genes via TF-SNVs varied per cell type (Fig. 2g). Myeloid and haematopoetic progenitor cells had the highest number of genes regulated by TF-SNVs, although the variation was quite large. Surprisingly, many genes (N = 503) were identified via TF-SNVs in different cell types and without a strong indication of

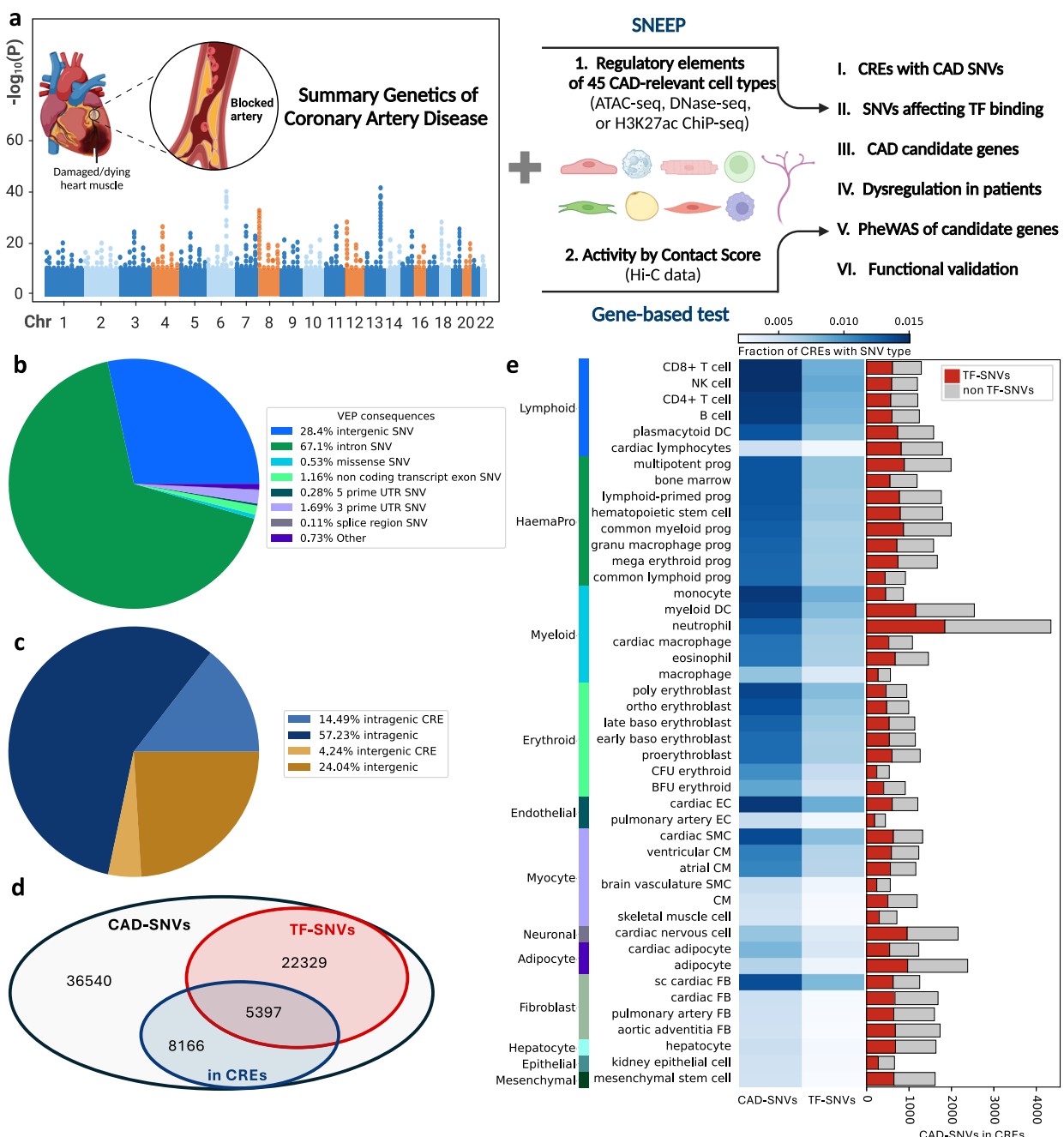

**Fig. 1 | Analysis outline and cell type-specific annotation of single nucleotide variants (SNVs) associated with coronary artery diseases (CAD). a** Overview of the project. GWAS of CAD[1] was combined with epigenome data and predicted cis-regulatory element (CRE) to gene interactions for 45 cell types. Candidate CAD genes were identified with a tool to annotate transcription factor (TF) SNVs (SNEEP[14]) and a gene-based association test (GATES[13]). The results were supplemented with data of CAD patients, PheWAS analyses and the functional validation of a candidate CAD gene. Created in BioRender. Chen, Z. (https://BioRender.com/t64i341). **b** Functional consequences of CAD-SNVs (p-value ≤ 2.52E-5 and SNVs in linkage disequilibrium) annotated with the Ensembl Variant Effect Predictor (VEP)[11]. Only one consequence is considered per variant. **c** Location of CAD-SNVs with respect to genes and CREs. **d** Venn diagram of the intersection of CAD-SNVs, TF-SNVs and SNVs located in CREs of any cell type. **e** Overlap of CAD-SNVs and TF-SNVs with CREs across cell types and grouped by lineage. Lineages sorted by average fraction of CREs with CAD-SNVs. NK cell natural killer cell, DC dendritic cell, BFU erythroid burst-forming unit erythroid, CFU erythroid colony-forming unit erythroid, CM cardiomyocyte, EC endothelial cell, SMC skeletal muscle cell, FB fibroblast. Source data are provided as a Source Data file.

genetic linkage, suggesting a range of different regulatory CAD-SNVs linked to the same gene (Fig. 2f).

Among the top 20 with the highest number of CREs containing TF-SNV, 10 were known CAD genes, 15 were also identified by the GWAS-eQTL colocalization analysis, and 14 were conserved in mice (Fig. 2h, and Supplementary Fig. 2). The top gene, *SMG6* (Telomerase-binding protein EST1A), had 21 CREs with TF-SNVs mostly in blood and immune cells. *SMG6* has been implicated with CAD through SNVs that affect post-transcriptional RNA methylation[26], by colocalization of eQTL data[21] and TF binding analysis[27]. To our knowledge, *B3GNT8* (Beta-1,3- N-Acetylglucosaminyltransferase 8) has not been implicated in CAD yet. However, B3GNT proteins have been associated with diabetes and processes of the immune system[28], which are risk factors for CAD. One of the three ncRNA genes with a high number of TF-SNV

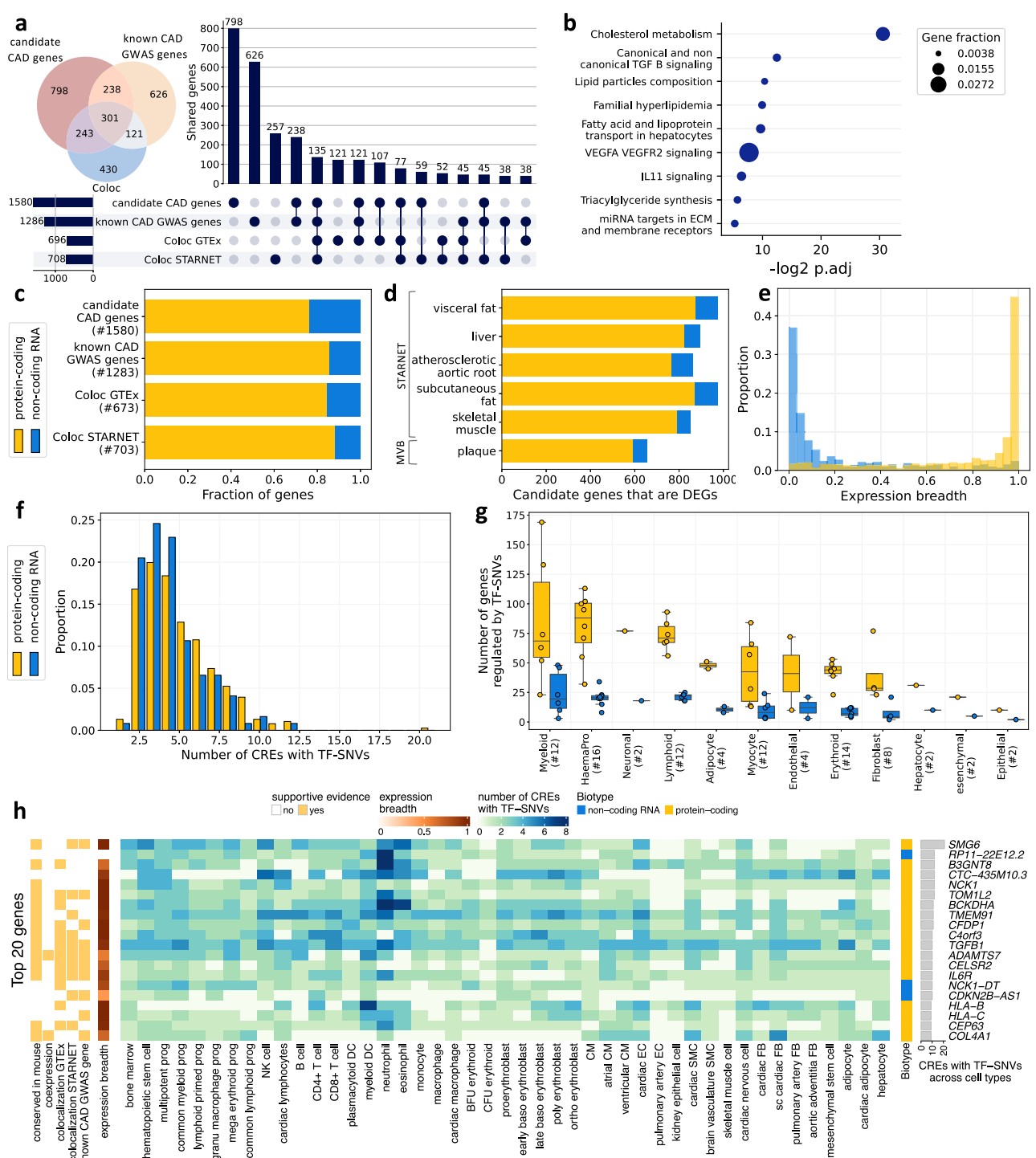

**Fig. 2 | Characterization of candidate coronary artery disease (CAD) genes.**
**a** Intersection of the candidate CAD genes, known CAD loci genes and genes found via eQTL colocalization analysis[16]. In the Venn diagram Coloc refers to the joint set of genes found via GTEx[17] and STARNET[18]. **b** GO term enrichment of the candidate CAD genes. Shown are selected terms from the top 20 enriched terms of the WikiPathways database (adjusted p-values from g:Profiler[77], without redundant terms and for Familial hyperlipidemia the adjusted p-value across all five types was averaged). **c** Fraction of protein-coding and non-coding RNA genes among the gene sets from (**a**). **d** Number of candidate genes that are differentially expressed (DEG, FDR ≤ 5% and absolute log2FC ≥ 0.3) across tissues from STARNET[18] and atherosclerotic plaque from the Munich Vascular Biobank (MVB)[23]. **e** Expression breadth of the candidate genes, separated by gene biotype. **f** Number of cis-regulatory

elements (CREs) per gene that has a transcription factor single nucleotide variant (TF-SNV), separated by gene biotype. Only genes found via TF-SNVs are shown (N = 503). The CREs with TF-SNV were merged across all cell types in which a gene had a CRE at its promoter and thus was considered active. **g** Number of candidate CAD genes identified via TF-SNVs grouped by lineage and separated by gene biotype (boxplots formed by the cell types assigned to a lineage, center line median, boxlimits inter-quartile range, whiskers up to 1.5x inter-quartile range). **h** Heatmap of the top 20 genes ranked by the number of merged CREs with TF-SNVs (as described for (**f**)). Additional evidence is added per gene (Supplementary Data 5). Per cell type the number of CREs with TF-SNVs is set to zero if the gene is not considered active in the cell type. Source data are provided as a Source Data file.

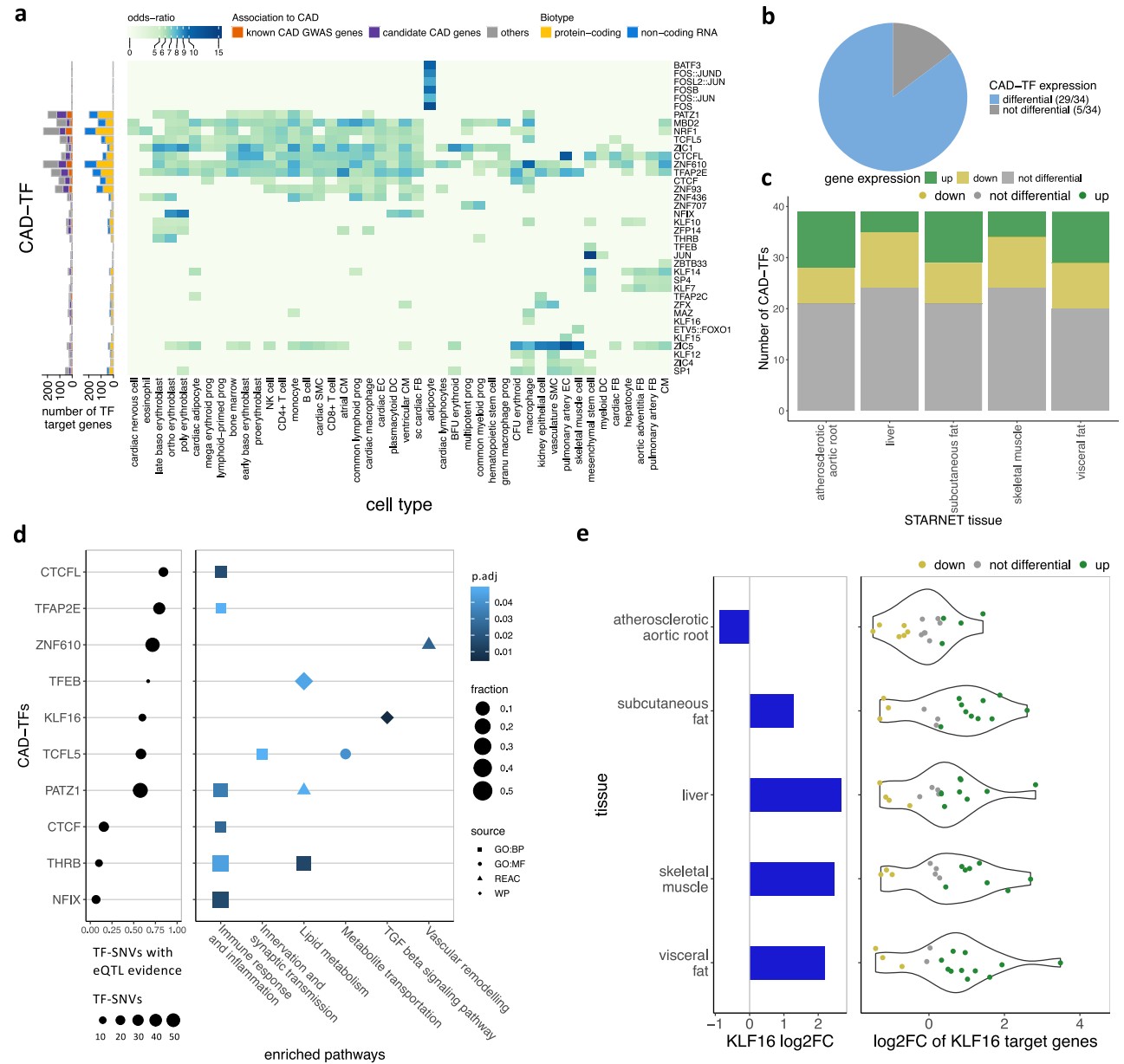

**Fig. 3 | Coronary artery disease (CAD) risk alleles affect transcription factor (TF) binding. a** Cell type-specific enrichment of CAD-TFs that are affected by binding to TF single nucleotide variants (SNVs). Each TF regulates a different number of target genes. By the stacked barplots (left) the fraction of target genes associated with CAD and the biotypes are visualized. **b, c** Analysis of differential expression of CAD-TFs (adjusted p-value from edgeR[80] of ≤ 0.05 and an absolute log2FC ≥ 0.3) in all (**b**) or individual STARNET tissues[18] (**c**). **d** CAD-TFs with eQTL support (GTEx[17]) and their associated biological processes and pathways (adjusted p-values from g:Profiler[77]). **e** Analysis of TF gene deregulation (log2 fold change, barplot) and expression deregulation of associated TF target genes in different STARNET tissues (log2 fold change, violin scatter) for KLF16. Genes with an adjusted $p$ value of ≤ 0.05 and an absolute log2FC ≥ 0.3 were defined as differential up- or downregulated (indicated in yellow and green). Source data are provided as a Source Data file.

containing CREs is *CDKN2B-AS1*. *CDKN2B-AS1* is located in the first ever identified and still strongest CAD GWAS locus, the 9p21 locus[29–31]. The candidate gene(s) at this locus and their mechanisms of action for CAD have been challenging to investigate[19,20]. We found *CDKN2B-AS1* as a candidate causal gene at this locus with 11 TF-SNVs in 10 different CREs which are mainly active in immune cells (Supplementary Fig. 3, and Supplementary Data 16).

### Genetic signals pinpoint CAD-related transcription factors
As a large subset of CAD-SNVs appears to affect TF binding (Fig. 1d), we further studied the TFs involved. The SNEEP-TF test was used to

discover TFs that are more often affected by a CAD-SNV than observed by chance (odds ratio ≥ 5, Methods). This analysis was done for each of the 45 cell types of epigenome samples and revealed 38 TFs (CAD-TFs), whose binding was changed by risk alleles of CAD in at least one cell type (Fig. 3a, and Supplementary Data 7). Among the predicted CAD-TF target genes are protein-coding and ncRNA genes and several are either known CAD GWAS genes or candidate CAD genes. The predicted binding of 26 TFs was affected in more than one cell type. Notably, the binding of the AP-1 family TFs was only enriched for adipocytes, including BATF3, FOSB, and FOS, and the dimers FOS::JUND, FOSL2::JUN, and FOS::JUN. While AP-1 family TFs fulfill a variety of

functions, their members have been specifically shown to be involved in differentiation and apoptosis in murine adipocytes[32]. Zhao et al.[33] found that the TFs of the AP-1 family regulate the known CAD genes *SMAD3* and *CDKN2B-AS1*, which are also in our CAD candidate gene set. These data underline the role of adipocytes or adipose tissues in CAD.

Further analysis using transcriptome data of disease-relevant tissues from controls and CAD patients of the STARNET project[18] revealed that the majority of CAD-TFs ($N = 29$) were differentially expressed in investigated tissues (Fig. 3b, and Supplementary Data 8). Within each tissue, an average of 44.7% of the CAD-TFs exhibited differential expression (Fig. 3c). CAD-SNVs regulating TF binding were also identified as eQTLs in CAD-relevant tissues, constituting independent evidence of disease-relevant TFs. The TFs CTCFL, TFAP2E, and ZNF610 had the highest eQTL support and the eQTL genes were involved in pathways for CAD such as *immune response and inflammation*, *lipid metabolism*, *TGF beta signaling*, and *vascular remodeling pathway* (Fig. 3d, and Supplementary Data 9, 10). Interestingly, the eQTL genes of CAD-SNVs regulating TCFL5 binding were enriched for *innervation and synaptic transmission*, a newly discovered pathway for CAD[12]. Furthermore, we observed that several TFs were up- or down-regulated in atherosclerotic aortic root tissue from STARNET, while showing the opposite regulation in fat, liver and skeletal muscle. Consistently, we also observed tissue-specific changes in the expression of their target genes. An example for a TF with such expression changes across patient tissues was KLF16 (Fig. 3e). It has been shown that KLF16 itself, as well as other TFs of the Krüppel-like family, are involved in CVDs and fulfill various functions in the cardiovascular system[34–37]. Similar tissue-specific effects of other TFs were observed as well, including PATZ1, BHLHA15, KLF7 and ZNF93 (Supplementary Fig. 4).

## Candidate genes associated with cardiovascular risk factors

To investigate the relevance of the identified CAD candidate genes, we performed the phenome-wide association study (PheWAS) on the tissue eQTLs of the 1580 candidate genes (Supplementary Fig. 5a). To do so in an unbiased manner, we started from the list of our candidate genes without adding any other information from the previous analyses. We then colocalized external eQTL data of our genes with GWAS data of relevant traits. The PheWAS datasets were obtained from the Common Metabolic Diseases Knowledge Portal and included the latest GWAS summary genetic statistics of 29 traits of CAD risk factors. We extracted the eQTLs from CAD-relevant tissue transcriptomes from GTEx[17] and STARNET[18]. Based on the probability of PheWAS-eQTL colocalization (Methods), we identified the phenotypic traits bridging our candidate genes and CAD. In total, 1173 candidate genes, including 1010 protein-coding and 163 non-coding genes, showed significant colocalization[16] (PPH4 ≥ 0.80) with at least one GWAS trait in a relevant tissue type (Supplementary Data 11, 12).

Overall, CAD candidate genes demonstrated the highest frequency of associations with inflammatory biomarkers (mainly immune cell counts) followed by lipid levels (mainly cholesterol and triglyceride (TG) levels), both of which involved over 800 genes (Fig. 4a). Artery and adipose tissues, which play essential roles in vascular and metabolic functions, exhibited the highest number of genes with significant colocalizations, with 811 and 761 prioritized genes, respectively (Fig. 4a). By comparison, tibial nerve, skeletal muscle, and blood revealed nearly equivalent gene counts with correspondingly 676, 642, and 639 genes. Interestingly, 35 genes showed significant PheWAS-eQTL colocalization signals in kidney cortex tissue, in line with the fact that chronic kidney disease is a risk factor for CAD[38]. The distribution of colocalizations across tissues does not match the general gene expression pattern, indicating that the PheWAS analysis finds independent regulatory effects (Supplementary Fig. 5b). For the novel CAD candidate genes (Fig. 4b), we observed similar patterns in comparison

to all prioritized CAD genes and known genes (Fig. 4a, and Supplementary Fig. 5c), again, suggesting the reliability of the novel genes.

For the top novel candidate genes ranked by their association strength with PheWAS traits, both protein-coding and ncRNA genes were predominantly associated with inflammatory biomarkers and lipid levels (Fig. 4c, d). Our data indicated that ncRNAs could contribute to CAD independent of protein-coding genes, especially for the intergenic ncRNAs, such as LINC01132. LINC01132 was uniquely associated with lipid levels. However, none of the top ncRNA genes showed conservation in mice. Interestingly, among the 163 ncRNA genes with PheWAS-eQTL colocalization signals, six were conserved in mice (Fig. 4e).

## *IQCH-AS1* contributes to CAD via obesity-related phenotypes

To further validate our novel findings, we conducted a biological validation on a novel CAD candidate gene, encoding a conserved lncRNA, *IQCH-AS1*. PheWAS-eQTL colocalization analysis suggested *IQCH-AS1* to be most significantly associated with obesity-related phenotypes including body mass index (BMI) and waist-hip ratio (WHR) (Fig. 4e). The GWAS signals of BMI and WHR were colocalized with *IQCH-AS1* eQTLs from adipose tissue, demonstrating that the roles of *IQCH-AS1* in this tissue might contribute to obesity and therefore increase the risk of CAD (Fig. 4f, and Supplementary Fig. 5f). We, therefore, studied the *IQCH-AS1* functions in human Simpson-Golabi-Behmel Syndrome (SGBS) preadipocytes and adipocytes. We first generated *IQCH-AS1* knockout (KO) SGBS preadipocyte lines by a dual CRISPR targeting strategy, which used two gene-specific single guided (sg) RNAs to target the shared exon of the major *IQCH-AS1* isoforms *IQCH-AS1*. The dual CRISPR excised 48 bp of the exon 13 and dramatically reduced the RNA level of *IQCH-AS1* compared to the scrambled control line (Fig. 5a, b). The mRNA and protein levels of the host gene *IQCH* were not affected (Supplementary Fig. 7a, b). By BrdU-based proliferation assay, we observed reduced proliferation of *IQCH-AS1*-KO preadipocytes which might indirectly lead to hypertrophy of existing adipocytes in hyperlipidemic conditions due to reduced source of new adipocytes (Fig. 5c). *IQCH-AS1*-KO preadipocytes had diminished differentiation efficiency indicated by less PPARG expression in the nuclei (Fig. 5d, and Supplementary Fig. 6). Therefore the corresponding differentiated adipocytes showed less accumulation of TGs (Fig. 5e), which could lead to TG increase in the circulation due to diminished capacity of lipid storage by adipocytes. Furthermore, *IQCH-AS1*-KO adipocytes released more proinflammatory cytokines including pentraxin 3[39] and IL-18[40], but fewer anti-inflammatory cytokines including IGFBP-3[41], Cripto-1[42,43], VEGF[44] and IL-10[45] (Fig. 5f). Our data from *IQCH-AS1*-KO preadipocytes and adipocytes indicates the gene to be protective for CAD. Indeed, *IQCH-AS1* was downregulated in SAT and VAT of atherosclerosis patients from STARNET[18] cohorts compared to controls (Fig. 5g). Its host gene *IQCH* showed similar downregulation (Supplementary Fig. 7c, d). Furthermore, the BMI- and WHR- increasing alleles at this locus were associated with reduced *IQCH-AS1* expression (Fig. 5h, and Supplementary Fig. 5g). The biological experiments suggest the CVD relevance of our novel gene, *IQCH-AS1*, which further supports the reliability of our epigenetics-GWAS analysis.

## Discussion

Our current work interpreted population genetics of CAD using epigenetics of 45 disease-relevant cell types. We showed that on average 1.04% of CREs from the cell types contained CAD-SNVs and ~50% of CAD-SNVs were located within CREs or predicted TF binding sites of the 45 cell types (Fig. 1). By GATES test and SNEEP analysis we identified 1580 candidate causal genes for the disease, including 1208 protein-coding and 372 ncRNA genes. CAD-associated ncRNA gene showed better cell type specificity than the protein-coding counterpart (Fig. 2e). 792 of our candidate genes were also found by published

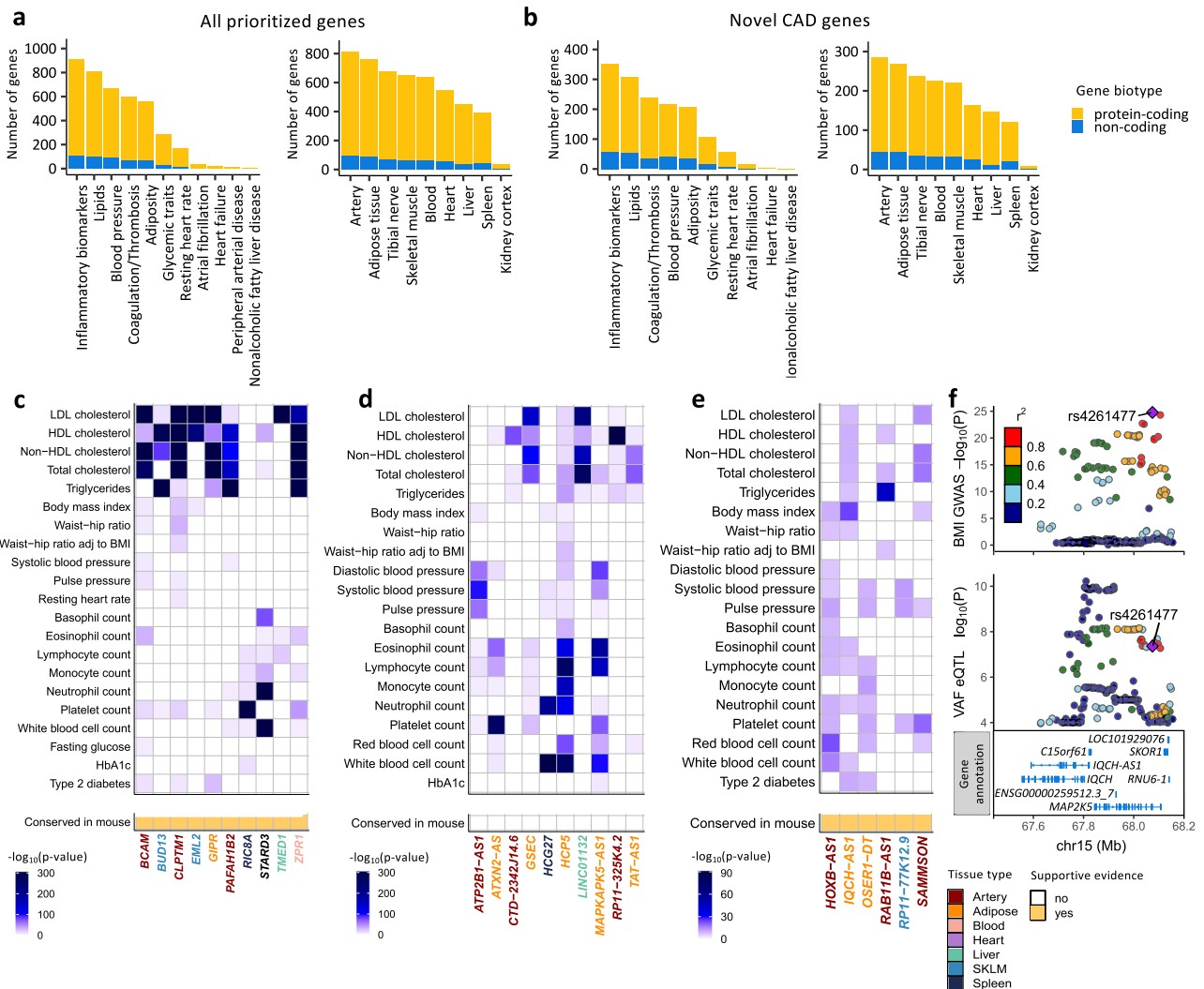

**Fig. 4 | PheWAS (Phenome-wide association study)-eQTL colocalization analysis on prioritized genes. a, b** Number of protein-coding and non-coding genes with significant colocalized GWAS-eQTL signals across phenotypes and tissue types among all prioritized (**a**) and novel coronary artery disease (CAD) genes (**b**). **c, d** Top 10 CAD-novel protein-coding (**c**) and non-coding (**d**) genes with the strongest eQTL effects on CAD-related phenotypes. The lowest p-value per gene

from the phenotype GWAS variants with colocalization is shown. **e** Phenotypic association of CAD-novel non-coding genes showed conservation in mouse. **f** *IQCH-AS1* locus displaying colocalization of the body mass index (BMI) GWAS with eQTL signals of visceral abdominal fat (VAF), PPH4 = 0.89. Source data are provided as a Source Data file.

studies or eQTL-based methods[1–3,46] and 798 were novel (Fig. 2). 542 out of 1,580 genes were regulated by the 38 TFs whose bindings were regulated by CAD-SNVs (Fig. 3). 81% of the candidate genes were differentially expressed in RNA data of patient cohorts. Based on PheWAS analysis, disease-relevant cellular phenotypes related to the candidate genes could be explored in the selected cell types. Both the novel protein-coding and ncRNA candidate genes showed similar PheWAS-eQTL colocalization patterns as the known candidates in terms of the associated traits and eQTL tissue types (Fig. 4), suggesting the reliability of our novel CAD candidate genes. Our biological validation on one of the novel candidate lncRNA genes, *IQCH-AS1*, indicated that the loss of the lncRNA could lead to detrimental adipose function, which was in line with its association with BMI and WHR, its downregulation in adipose tissue of CAD patients and the eQTL correlation in the same tissues (Fig. 5). Knockout of *IQCH-AS1* affected preadipocyte proliferation and lipid accumulation and therefore could play roles in obesity and CAD. Further molecular mechanisms shall be examined to explore the therapeutic potential of this RNA.

Our work identified more candidate genes ($N = 1580$) for CAD than the published studies ($N = 1286$) or eQTL-based methods

($N = 1169$)[1–3,10,46]. Both the datasets and the statistical methodologies differ our work from others, which could lead to improved discovery. First, we included epigenetic data from 45 cardiovascular cell types, representing the most diverse cell groups investigated in CAD genetic studies. Second, our analyses also appear to be more sensitive than the GWAS-eQTL colocalization using GTEx and STARNET datasets, which identified 1095 genes of nine CAD-relevant tissues in total. Given these eQTLs were mapped using tissue-level data, the eQTL signals with small effect sizes due to high cell specificity might be obscured in the bulk transcriptomic profiling. While future systematic single-cell eQTL datasets could make up for the drawback, our integration of cell type epigenetics has unveiled CAD candidate genes and CAD-TFs with high cellular specificity, such as *FLT1* in dendritic cells and *SLC22A3* or the AP-1 family TFs in adipocytes. Third, we used two complementary strategies, the GATES test and the SNEEP analysis, to prioritize CAD candidate genes. The GATES test summarizes the significance of all SNVs in the gene region and thus considers SNVs in coding and non-coding genic regions. The SNEEP analysis combines the prediction of TF-SNVs with the occurrence in cell type-specific regulatory elements and is thus focused on the non-coding part of the (epi)genome. That

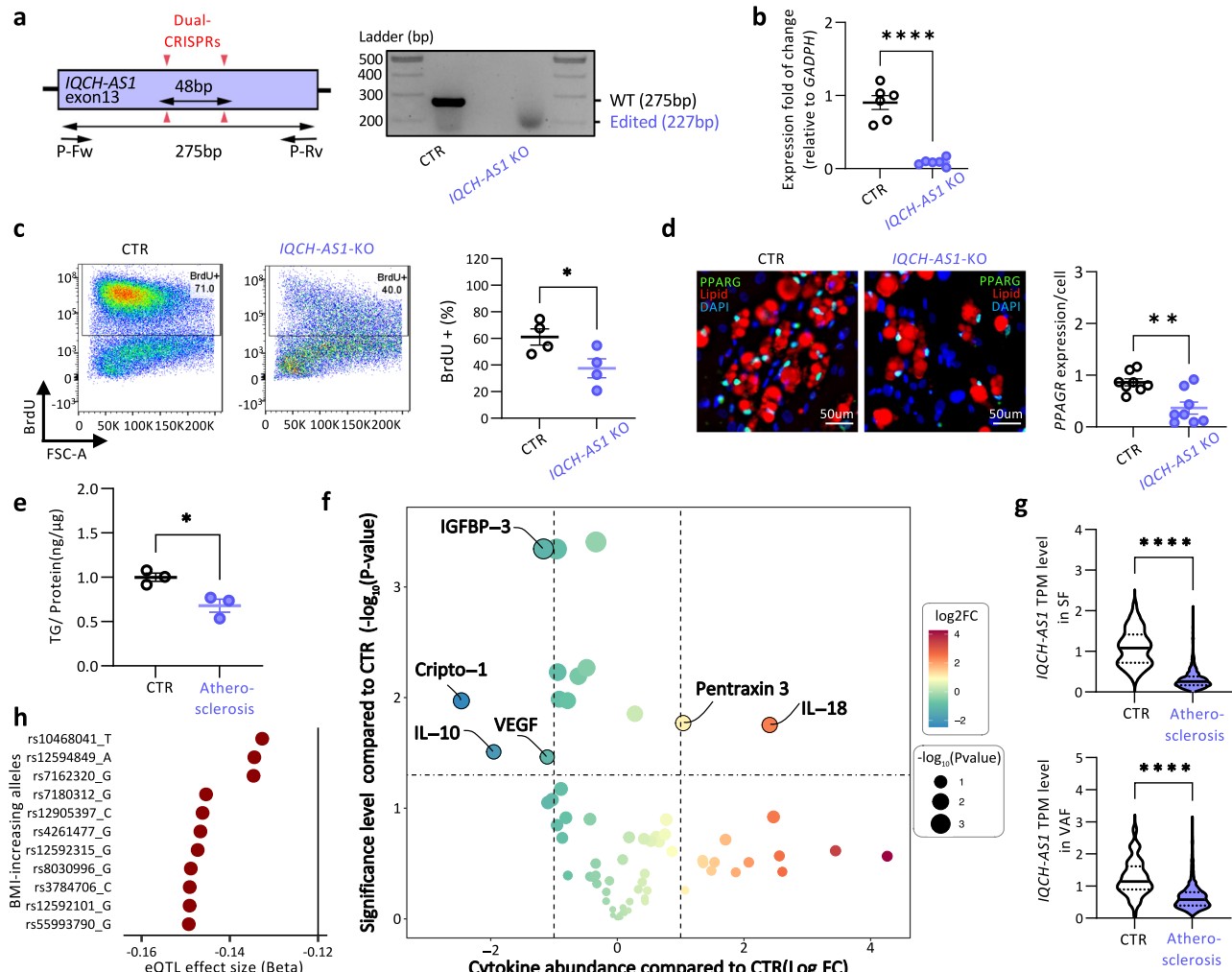

**Fig. 5 | Atherogenic phenotypes induced by CRISPR-based knockout (KO) of IQCH-AS1. a** Dual CRISPR-KO of *IQCH-AS1* in SGBS preadipocytes perturbed the IQCH-AS1 locus. The experiment was independently repeated with *n* = 5 biological replicates, all showing similar results. **b** *IQCH-AS1* RNA levels were significantly reduced, measured by qRT-PCR. Data are presented as mean ± SEM from *n* = 6 independent biological replicates; unpaired two-tailed t tests comparing each sample with Control (CTR) (*P* < 0.0001). **c** Loss of *IQCH-AS1* reduced the proliferation of SGBS preadipocytes, as indicated by decreased numbers of BrdU-positive cells. Cells were analyzed by forward scatter (FSC-A) versus BrdU intensity to assess proliferation. Data are presented as mean ± SEM from *n* = 4 independent biological replicates; unpaired two-tailed t tests comparing each sample with Control (*P* = 0.0465). **d, e** *IQCH-AS1-KO* adipocytes exhibited reduced PPARG expression (**d**) and decreased triglyceride (TG) accumulation (**e**). PPARG was detected using a protein-specific antibody (green), lipid droplets were stained with HCS LipidTOX™ Red neutral lipid stain (red), and nuclei were counterstained with DAPI (blue) (Supplementary Fig. 6). Data in (**d**) are presented as mean ± SEM from *n* = 8 independent biological replicates, unpaired two-tailed t tests comparing each sample with Control (*P* < 0.0023). Data in (**e**) are presented as mean ± SEM from *n* = 3 independent biological replicates per group, unpaired two-tailed t tests comparing each sample with Control (*P* = 0.0213). **f** Proteome Profiler analysis of conditioned media revealed altered cytokine secretion from *IQCH-AS1-KO* adipocytes. The *x*-axis represents log2 fold change (log2FC), with genes on the right (log2FC > 0) up regulated and genes on the left (log2FC < 0) down regulated. The *y*-axis represents -log10(*p* value). Genes with absolute log2FC > 1 and *p* < 0.05 were considered significantly differentially expressed. The experiment was independently repeated with *n* = 4 biological replicates. Differential expression was calculated using DESeq2. **g** *IQCH-AS1* expression was significantly reduced in subcutaneous fat (SF) and visceral abdominal fat (VAF) from patients with atherosclerosis compared with controls. Data are presented as mean ± SEM from *n* = 568 patients and *n* = 92 controls for SF, and *n* = 531 patients and *n* = 103 controls for VAF; unpaired two-tailed t tests comparing each sample with Control (*P* < 0.0001 for both SF and VAF). Box plots show the median (center line), 25th–75th percentiles (box), and min–max values (whiskers). **h** Effect of BMI-increasing alleles of *IQCH-AS1* eQTLs in VAF on *IQCH-AS1* expression. The lead BMI-associated SNP (rs4261477) and ten SNPs in high linkage disequilibrium (R² > 0.8) were included from Fig. 4F. *,*p* ≤ 0.05. ****, *p* ≤ 0.0001. Source data are provided as a Source Data file.

these tests draw their power from different statistical principles explains their complementarity. It is noteworthy that the way the prioritization is done using SNEEP is more stringent than in a previous application[47]. A controlled randomization approach suggested the use of the occurrence of at least *two* regulatory elements with TF-SNVs not in LD as a more stringent threshold to avoid false positives (Supplementary Fig. 1e). Fourth, genetically regulated non-coding RNA genes were systematically explored, which increased our discoveries. Among

our candidate CAD genes, 23.5% were non-coding RNA genes, representing the highest ratio compared to reported candidate genes and genes found via GWAS-eQTL colocalization analysis.

In our study, microRNAs were prioritized as candidate genes for CAD by genetic analysis. This is not possible with the population transcriptome-based method, given the existing genotyped transcriptome datasets, such as GTEx and STARNET datasets, excluded short RNA from the profiling. Interestingly, microRNA132 (MIR132), a

therapeutic target in a Phase 2 clinical trial (NCT05350969) for heart failure[22], was also exclusively underlined by our analysis (Supplementary Fig. 2b). Our analysis suggested MIR132 might also play a role in CAD. Indeed, the CREs containing TF-SNVs for this ncRNA gene were not only from cardiac cells but also immune cells. In addition, 361 lncRNA genes were prioritized, among them only 14 were conserved between humans and mice, which permits biological studies and therapeutic explorations both in vitro and in vivo. In our functional validation experiments, we demonstrated the novel ncRNA, *IQCH-AS1*, indeed was involved in CAD-relevant cellular functions.

Limitations exist in our study. Our analysis can not differentiate the role of the microRNA genes from the host genes. We therefore excluded microRNA genes in intragenic regions from our final candidate gene list. Further, the collection of epigenome data consists of different assays (DNase-seq, ATAC-seq, H3K27ac ChIP-seq) which can differ in the regulatory regions they uncover and in turn determine which SNVs can be found via epigenome analysis. However, a uniformly assayed collection for such a large collection of CAD-relevant cell types is not available. The predictions for CRE-gene interactions allow one CRE to be linked to multiple genes. In these cases, we assume all target genes to be affected, although the effect might be limited to individual genes. We analyzed TFs whose binding sites are affected independently, despite TFs also functioning in a combinatorial manner[48,49]. Systematic incorporation of TF cooperativity would require knowledge of all possible TF interactions, which is not available. The prediction of TF-SNVs is based on TF binding motifs, which are imperfect for annotating binding sites, but enable us to analyze such a large collection of cell types and TFs. Another point to consider is that the GWAS and LD data are not from the same population. While both are of majorly European ancestry, we might over- or underestimate the linkage of SNVs from other ancestries. Efforts can be invested to explore ancestry-specific epigenetic regulatory circuitry of CAD loci after a large growth of GWAS sample size from other populations, such as African, Hispanic and Asian.

Nevertheless, our study created an inventory of CAD candidate genes regulated by genetically affected epigenetics in a cell type-specific fashion. We prioritized novel candidate genes beyond existing methods, including protein-, lncRNA- and microRNA-coding genes. We demonstrated the significance of TFs in genetically regulated CAD risk. The results could lay the foundation for further biological investigation and the therapeutic exploration of CAD candidate genes.

## Methods

### Collection of enhancer-gene interactions

Epigenome data indicating CREs (DNase-seq, ATAC-seq, and H3K27ac ChIP-seq) were collected for 45 cell types related to CAD (Supplementary Data 1, 2). For each cell type the set of candidate CREs was defined as the peaks of the epigenome data. Most studies already provided peak annotations. For the scATAC-seq data of Hocker et al.[50], the bigWig files of the cell types were converted to BedGraph with UCSC's bigWig-ToBedGraph executable[51], followed by MACS2's bdgpeakcall function (v2.2.7.1)[52] with a minimum length of 100 bp and signal cut-off of 0.004. The activity of the peaks was afterward assessed by taking the average signal of the bigWig-files. For the peak files of all cell types, replicates were merged and if the data was in genome version hg19 they were lifted to hg38 with a Python implementation of UCSC's liftOver tool (v1.1.13)[53]. Handling of bed-files was done with pybedtools (v0.8.1)[54,55]. To predict CRE-gene interactions for each cell type the generalized ABC-score from the STARE framework (v1.0.4)[15] was used, using the average Hi-C data from Gschwind et al.[56] as chromatin contacts. The maximum distance between CRE and a gene's TSS was set to 2.5 MB. CREs overlapping regions known to accumulate anomalous signals from sequencing experiments were excluded[57,58]. All interactions surpassing a gABC-score of 0.02 were considered relevant. The command was as follows:

```
STARE_ABCpp -b <peak_file> -n <activity_column> -a gencode.-
v38.annotation.gtf -w 5000000 -f <ENCFF134PUN_avgHiC_hg38/> -k
5000 -t 0.02 -x hg38-blacklist.v2.bed -o <output_path>.
```

### Identification of TF-SNVs and CAD-TFs

The GWAS summary statistic was downloaded from a recently published study for coronary artery disease[1]. Similar to Aragam et al.[1], a 1% FDR cutoff was applied to extract those SNVs significantly associated with CAD ($p \leq 2.52\text{E-}5$). SNVs in linkage disequilibrium (LD) were determined based on the pre-computed dataset from SNiPA[59], where the European cohort and an LD threshold $\geq 0.8$ were used. The lead and proxy SNVs were lifted to hg38 using the Python package liftover[53]. The resulting 72,432 SNVs are used in the following analyses (Supplementary Data 3).

The SNEEP software (version v1.0)[14] was applied for the identification of TF-SNVs (Supplementary Data 4), disease genes, and disease-associated TFs. Based on the beta coefficient given in the summary statistic (denoted as beta) the alleles were ordered in such a way that SNEEP compares the CAD-risk allele against the non-risk allele.

TF-SNVs were predicted for each of the 45 cell types individually. From the 72,432 lead and proxy SNVs, those not overlapping with the cell type-specific enhancer-gene interactions were excluded. Further, these interactions were used to associate the TF-SNVs to cell type-specific target genes. As TF motifs, 817 non-redundant human motifs from JASPAR (version 2022)[60], HOCOMOCO[61], and Kellis ENCODE motif database[62] were used (https://github.com/SchulzLab/SNEEP/).

Further, 500 background analyses were performed based on randomly sampled SNVs. Whenever possible the random SNVs were derived with SNPsnap[63,64] to ensure random SNVs were most similar to the original ones in terms of minor allele frequency (MAF), distance to the nearest gene, gene density, and SNVs in LD (the used SNPsnap reimplementation from A. Abraham is provided in our GitHub repository and required data is available in our Zenodo repository). However, for less than 5% of all lead and proxy SNVs, SNPsnap failed since the LD structure of the SNV could not be derived. In such a case the corresponding random SNV with a similar MAF was sampled from the dbSNP database[65]. The following SNEEP command was performed per cell type:

```
differentialBindingAffinity_multipleSNPs -p 0.5 -c 0.001 -f <cell-
typeSpecificEnhancerGeneInteractions_merged> -r <celltypeSpecifi-
cEnhancerGeneInteractions> -g <geneId_geneName.txt> -j 500 -l
 <combined_motif_set> <snvFile> <hg38.fa> <
estimatedScalesPerMotif_1.9.txt>.
```

To explore genes significantly affected by the analyzed SNVs, the randomly sampled SNVs were linked to target genes using the cell type-specific enhancer-gene interactions. Next, it was counted per gene how many non-LD TF-SNVs in different CREs were observed in the random data. Based on the resulting cell type-specific background distribution over all genes, a FDR corrected $p$ value was derived. Overall cell types observing at least two non-LD TF-SNVs were significant (FDR $\leq 0.01$), suggesting that the associated target genes are highly affected by the CAD-SNVs (Supplementary Fig. 1e).

Based on the background analysis, 38 cell type-specific TFs were derived that are more often affected by the analyzed SNVs than expected on a random background control (Supplementary Data 7). The analysis was done as described by Baumgarten et al.[14] (see Star Methods Section, eQTL analysis of this paper): for each TF an odds ratio was computed given the numbers of how often a TF's binding site was significantly affected by the analyzed SNVs (TFcount) in comparison to the random SNVs (bgCount). We defined the odds-ratio for each TF as $\text{odds} - \text{ratio(TF)} = \frac{\alpha/(1-\alpha)}{\beta/(1-\beta)}$, where $\alpha = \frac{TFcount}{\#SNVs}$, $\beta = \frac{bgCount}{\#SNVs}$ and #SNVs are the number of CAD-SNVs. CAD-TFs are defined as those TFs with more than five TF-SNVs, an odds ratio $\geq 5$ in at least one of

the analyzed cell types, and an open promoter region in the same cell type.

## Functional consequences of CAD-SNVs with VEP

For the functional annotation of variants the Variant Effect Predictor (VEP) from Ensembl (v111.0)[11] was used with the 'pick' option to get one consequence per variant. The distance to consider up- or downstream variants was set to zero. Categorical colors were taken from the colorcet Python package, based on Glasbey et al.[66].

## Gene-based test

In addition to SNEEP the gene-based association test GATES[13] was applied to identify CAD-associated genes. For each gene, all SNVs from the CAD GWAS summary statistic[1] (lifted to hg38) that overlap the genes' bodies were considered. The gene body was defined as annotated in the GENCODE's[67] v38 gtf-annotation including exons and introns. To determine the LD structure for all pairs of SNVs associated with a gene, PLINKs[68] functionality (http://pngu.mgh.harvard.edu/purcell/plink/, version 1.07) to look up pair-wise LD correlation (R2) was used:

plink-1.07-x86_64/plink --noweb --bfile alkesgroup.broadinstitute.org/LDSCORE/GRCh38/plink_files/1000 G.Eur.hg38.<chromosome> --snps <list_rsIDs> --r2 --matrix --ld-snp-list <gene> --out <gene>.txt.

For the LD score data we downloaded the 1000 Genomes European ancestry LD files from (https://alkesgroup.broadinstitute.org/LDSCORE/GRCh38/) in March 2022. The SNVs for which the pair-wise LD score could not be derived were excluded. Next, GATES was computed for each gene with SNVs in the gene body. The resulting $p$ values were FDR-corrected with the Benjamini-Hochberg procedure[69]. The implementation for GATES was taken from the R package COMBAT[70] (version 0.0.4) and slightly adapted.

## Defining candidate CAD genes and adding further evidence

We combined the results of the TF-SNV analysis and the gene-based association test to define a joint set of candidate CAD genes. All genes from the SNEEP analysis that had at least two non-LD TF-SNVs in different CREs in at least one cell type in which the gene was also active were considered candidate CAD genes. A gene was considered active in a cell type if a peak called on the epigenetic signal overlapped any of the gene's promoters (± 200 bp, Supplementary Data 5). For the GATES test, a gene had to have at least 10 SNVs in its gene body, be open in any cell type, and have an FDR ≤ 2.5%. Specific genes were excluded from all gene sets and not considered as potential candidate CAD genes, all based on GENCODE's[67] v38 gene annotation: pseudogenes, genes that are not yet experimentally confirmed (labeled as TEC), miRNAs that are located completely in the gene body of other genes on the same strand and genes which share their 5' TSS with other genes. Overall, 17,732 genes from the annotation were excluded (Supplementary Data 5).

Additional evidence and data were gathered for the candidate CAD genes (Supplementary Data 5). It was checked whether the candidate genes were also identified in other types of analyses. A manually curated list of known CAD loci genes was generated, comprising 1,286 genes (after removing the excluded genes) and which is based on previously published work combining the following criteria: 1) the biological plausibility, 2) rare coding variant(s) associated with CAD, 3) likely pathogenic variant(s) relevant to CAD in ClinVar[5], 4) evidence from effective cardiovascular (CV) drug(s), 5) significant causality by Mendelian Randomization studies, 6) a protein-altering variant in high LD with the sentinel CAD variant, 7) expression quantitative trait loci (eQTLs) in a CAD-relevant tissue, and 8) a CV-relevant phenotype in

knockout mouse models[1,2,6]. Gene names were mapped to Ensembl IDs with the help of my MyGene.info[71–73].

Further, it was added whether a gene was also found via colocalization analysis with eQTL data from GTEx[17] and the STARNET studies[18] or among candidate CAD genes from a recent publication that focused on endothelial cells[10]. In addition, it was tested whether the candidate CAD genes are co-expressed with known CAD loci genes. To do so, a co-expression analysis was performed using gene expression profiles of 9662 GTEx RNA-seq samples[17]. For each candidate CAD gene, the top 100 co-expressed protein-coding genes were derived using the Spearman correlation coefficient as a similarity metric. Applying Fisher's exact test, 74 candidate CAD genes were significantly co-expressed with known CAD loci genes (FDR ≤ 0.05). The expression breadth was determined based on RNA-seq data from the IHEC EpiATLAS (1555 samples, https://ihec-epigenomes.org/epiatlas/data/) and calculated as the fraction of cell types/tissues ($N = 58$, metadata column 'harmonized sample ontology intermediate', metadata version 1.1) in which a gene had an expression of ≥ 0.5 transcripts per million. An expression breadth of 1 means that a gene is expressed in all cell types/tissues. If a gene is conserved in mouse was based on the annotated orthologues from BioMart[74] (confidence > 0). To also account for lncRNAs, a Reciprocal Best BLAST Hit[75] was performed (sequence version GRCm39 for mouse and GRCh38 for human). The human lncRNA sequences were aligned to the mouse genome (forward) and the process was repeated the other way around (reverse). If a human gene had a reciprocal best match without duplicate alignments, it was considered to be conserved in mice. Overlap of gene sets were visualized with UpSet plots[76]. All GO term enrichment tests were done with g:Profiler's Python package (v.1.0.0)[77]. For the enrichment test of the candidate CAD genes (Fig. 2b), all genes active in any cell type and not part of the set of excluded genes were used as background. As background for the TF target genes, all genes active in any cell type were used.

## Differential expression of genes in CAD patient data

Differential expression analysis using the Munich Vascular Biobank (MVB)[23] dataset was performed on 145 early and 57 advanced human carotid plaque lesions collected from patients undergoing carotid endarterectomy as previously published[24]. Raw sequencing data in FASTQ format were subjected to quality assessment using FastQC. Reads were aligned to the human reference genome (GRCh38) using the STAR aligner[78]. Genes with fewer than one count per million (CPM) reads across all samples were removed, resulting in 38,574 genes retained for further analysis. Sample-level quality control was performed by manual inspection of quality metrics in combination with automated outlier detection methods, including distance-based metrics, Kolmogorov–Smirnov tests, correlation analysis, and Hoeffding's D statistics. Raw read counts were normalized with trimmed mean of M-values (TMM) and transformed with voom from the limma package (v3.50.1), resulting in log2-counts per million with associated precision weights, which were then used for differential expression analysis via linear modeling with limma[79].

Differential expression profiles for the atherosclerotic aortic root, liver, subcutaneous fat, skeletal muscle, and visceral fat tissues between ~600 individuals with CAD and ~100 CAD-free controls were estimated using the edgeR[80] R package using the raw RNA-sequence data from the STARNET study[18,81] (dbGaP accession number phs001203.v1.p1) (Supplementary Data 8). Genes with an adjusted $p$ value of ≤ 0.05 and an absolute log2FC ≥ 0.3 were defined as differential. 34 out of 38 CAD-TFs were available in the STARNET data set (Fig. 3b, c). The expression of the target genes (log2FC) of TFs KLF16, KLF7, BHLHA15, ZNF93 and PATZ1 was extracted for the tissues in which the TF was differentially expressed (Fig. 3e and Supplementary Fig. 4).

## eQTL support for CAD-genes

The eQTL data for 49 tissues and cell types was downloaded from the GTEx[17] portal (version 8). Only the eQTLs linked to known CAD GWAS genes or candidate CAD genes were kept. Additionally, it was filtered whether the eQTL was found in a CAD-relevant tissue included in the GTEx data (aorta, coronary artery, mammary artery, tibial artery, visceral and subcutaneous adipose tissues, whole blood, heart atrial appendage and left ventricle, kidney cortex, liver, skeletal muscle, spleen, and tibial nerve). For each CAD-TF, the TF-SNVs from the cell types the CAD-TFs were identified in were gathered. The fraction of eQTLs supporting the TF-SNVs linked to target genes was computed (Fig. 3d right panel, Supplementary Data 9).

## GWAS sources and eQTL datasets for colocalization analysis

The 29 GWAS datasets of CAD-relevant traits or diseases were full-genome summary statistics from publicly available studies (Supplementary Data 13). eQTLs of artery (aorta, coronary, mammary, tibial), adipose tissues (visceral, subcutaneous), blood, heart (atrial appendage, left ventricle), kidney cortex, liver, skeletal muscle, spleen, and tibial nerve, were obtained from STARNET[18] and GTEx[17] v8 datasets. The STARNET dataset was obtained via collaboration and the GTEx dataset was accessed via the dbGaP platform.

## GWAS-eQTL colocalization analysis with COLOC

COLOC represents a Bayesian approach that estimates the posterior probabilities between a given GWAS signal and a given QTL[16,82–84]:

H0: Neither has a significant association in the region
H1: Only the GWAS trait has a significant association in the region
H2: Only the QTL trait has a significant association in the region
H3: Both GWAS and QTL traits have a significant association in the region, but the variants are different
H4: Both GWAS and QTL traits have a significant association and share the variants in the region

For each CAD candidate gene, we obtained eQTL SNVs from GTEx and STARNET studies with $p$ value < 0.01. These SNVs were used to test the colocalization with the GWAS associations of the 29 datasets. GWAS SNVs with p-value < 5e-8 were selected for the analysis. To estimate the posterior probability of GWAS-eQTL colocalization we used the coloc R package and ran the coloc.abf function. GWAS and eQTL datasets with the effect sizes, allele frequencies, and sample sizes were used as inputs. Significant colocalizations were defined by PPH4 ≥ 0.80, indicating shared association signal and variants between GWAS and eQTL (Supplementary Data 11, 12). Top genes were selected based on their GWAS association strength, with priority given to the lowest GWAS $p$ values and the highest number of colocalized traits.

## Cell culture and adipogenic differentiation

HEK293T cells (ATCC, USA) were maintained in high-glucose Dulbecco's Modified Eagle Medium (#11965092, ThermoFisher, Waltham, USA) supplemented with 10% fetal bovine serum (#S0615, Sigma Aldrich, St. Louis, Missouri, USA). Human Simpson-Golabi-Behmel Syndrome (SGBS) preadiocytes, a non-immortalized preadipocyte cell line, were provided by Dr. Daniel Tews and Prof. Martin Wabitsch, and cultured in DMEM/F-12 (#11330057, ThermoFisher, Waltham, USA) supplemented with 10% FBS (#S0615, Sigma Aldrich, St. Louis, Missouri, USA) and 100 U/l penicillin/streptomycin (#15140122, Thermo-Fisher, Waltham, USA). All cultures were incubated at 37 °C in a humidified atmosphere containing 5% CO₂. Medium was replaced every 48 h, and cells were subcultured when they reached ~90% confluence. Cell differentiation was performed as follows[85]: SGBS pre-adipocytes were cultured to confluence and induced to differentiate in serum- and albumin-free DMEM/F12 containing penicillin/streptomycin (100 U/mL), biotin (3.3 μM), pantothenate (1.7 μM), transferrin

(10 μg/mL), insulin (5 μM), cortisol (200 nM), triiodothyronine (0.2 nM), rosiglitazone (2 μM), IBMX (0.5 mM), and dexamethasone (12.5 nM). After 4 days, the medium was replaced with DMEM/F12 containing transferrin, insulin, cortisol, and triiodothyronine, yielding >90% adipogenic differentiation by day 14.

## CRISPR/Cas9 plasmids and virus infection

SgRNA sequences for lentiviral plasmid construction were designed using the online tool CHOPCHOP[86]. Two complementary oligonucleotides were synthesized: 5′-CACCG-[sgRNA sequence-3]′ and 5′-CAAA-[reverse complement of sgRNA]-C-3′. Two sgRNAs targeting the shared exon of all transcripts were delivered via lentivirus into SGBS-preadipocytes. Exon 13 of IQCH-AS1 was specifically targeted using a dual CRISPR strategy, resulting in a 48 bp frameshift deletion. For lentiviral production we followed our previous work[87]: sgRNAs cloned into LentiCRISPR v2 (#52961, Addgene, Watertown, USA) were co-transfected into HEK293T packaging cells with the packaging plasmids psPAX2 and the VSV-G envelope plasmid pMD2.G using FuGENE® HD (#E5911, Promega, Madison, USA) according to the manufacturer's instructions. Viral supernatants were collected 48–72 h after transfection, clarified by centrifugation, filtered through a 0.45 μm filter, and either used immediately or stored at −80 °C. For transduction of SGBS preadipocytes, cells were plated at ~50% confluence and exposed to lentiviral supernatants supplemented with 8 μg/mL polybrene to enhance infection efficiency. The cells were selected with 2 μg/mL puromycin for 5 days to remove non-infected cells. The positively targeted cells were then expanded in culture and subsequently passed for further analysis. Sequences of sgRNAs and primers are provided in Supplementary Data 14. All newly generated CRISPR plasmids and modified SGBS cell lines are available from the corresponding author upon reasonable request.

## Amplification PCR and Real-time PCR (RT-qPCR)

Total RNAs were isolated from cells using TRIzol reagent, and an amount of 3 μg of RNA was used in 10 μl reaction volume to digest DNA using the Maxima H Minus cDNA Synthesis Master Mix (#15606029, ThermoFisher, Waltham, USA) following the manufacturer's instructions. PCR amplification was performed with Q5 High-Fidelity 2X Master Mix (#M0492L, New England Biolabs, Massachusetts, USA) using 750 ng of cDNA template on a thermal cycler. The PCR products were electrophoresed using 1% agarose in 1×TBE buffer at 120 V for 45 min, with *GAPDH* as an internal control (Supplementary Data 14). For RT-qPCR, SYBR Green probes (#95074-012, VWR International, Radnor, Pennsylvania, USA) were used to detect the target genes and 60 ng cDNA was used as the template. *GAPDH* was used as an internal control. Reactions were performed on a ViiA 7 Real-Time PCR System (ThermoFisher, Waltham, USA). Expression levels were reported as 2-DCt values.

## Lipid extraction and triglycerides measurement

As described previously[87], the SGBS adipocytes were first washed twice with PBS, and then detached by scraping. Lipids were extracted using a chloroform mixture (2:1, v/v), followed by overnight drying in a fume hood. The dried lipid residue was resuspended in 100 μL of 1% Triton X-100 in absolute ethanol and incubated for 1 hour with constant rotation. After incubation, the suspension was dried in a Speed-Vac for 30 minutes, the residue was then resuspended in 100 μL of PBS with 1% Triton X-100. To measure lipid content, a 3 μL aliquot of the suspension was taken, and lipid triglycerides were measured using a triglyceride determination kit (#TR0100, Sigma Aldrich, St. Louis, USA).

## Immunofluorescence staining

Cells were washed with PBS and fixed in ice-cold methanol at -20 °C for 15 min. After PBS washing, cells were blocked for 1 h at room

temperature with PBS containing 5% m/v BSA. Cells were incubated overnight at 4 °C with rabbit anti-PPARgamma primary antibody (dilution 1:200, #2443S, Lot: 4, Cell Signaling Technology, Danvers, USA). After PBS washing, Donkey anti-rabbit lgG Highly Cross-Adsorbed Secondary Antibody (dilution 1:500, #a32790, Lot: YC363758, ThermoFisher, Waltham, USA) and HCS LipidTOXred neutral lipid stain (dilution 1:1000, #H34476, Lot: 2441365, ThermoFisher, Waltham, USA) were applied. Cells were then mounted with a DAPI mounting buffer (dilution 1:1000, #62248, Lot: YK4113231, Thermo-Fisher, Waltham, USA). Fluorescence signals were visualized by the THUNDER imaging system (Leica, Wetzlar, Germany).

### Cell proliferation assay
The BD Pharmingen™ APC BrDU Kit (#552598, BD Biosciences, Franklin Lakes, USA) was used following the manufacturer's protocol. Cells were treated with 10 µmol/L BrdU for 24 hours, then were collected, fixed, and permeabilized using the kit. BrdU incorporation was assessed with a BD LSRFortessa™ Cell Analyzer (BD Biosciences, Franklin Lakes, USA). The recorded events were gated using forward- (FSC-A) and side-scatter (SSC-A) adopted to the cell type. Then, single cells were gated, using FSC-A versus forward scatter height (FSC-H). Finally, within the single cell population, BrdU$^+$ population was identified. Unstained control samples were used as negative control of the fluorescence and to set the gate BrdU$^+$ cells. Gating was applied to exclude doublets and flow cytometry data were analyzed using FlowJo (version 10.8.2, RRID: SCR 008520).

### Proteome profiler human XL cytokine array
Cytokines released from control and IQCH-AS-KO adipocytes were analyzed using the Proteome Profiler Human XL Cytokine Array (R&D Systems, Minneapolis, MN, USA). Specifically, cells for the assay were cultured for 24 h in a serum-free medium. Mediums were first collected and enriched five times by vacuum centrifuge, then incubated with a biotinylated detection antibody cocktail on a nitrocellulose membrane. After washing, the membrane was incubated with streptavidin–horseradish peroxidase. Cytokine signals were detected with the ImageQuant 800 imaging system (Amersham Biosciences, Amersham, UK). Cytokine signal density was quantified using dot plot analysis in the ImageQuant TL software and then normalized by subtracting the background intensity. The normalized signal density was compared between control and *IQCH-AS1*-KO adipocytes.

### Protein extraction and immunoblotting
Cells were harvested and lysed on ice using RIPA buffer (#9806, Cell Signaling Technology, Danvers, Massachusetts, USA) supplemented with a protease and phosphatase inhibitor cocktail (#78447, Thermo Fisher Scientific, Waltham, Massachusetts, USA). Insoluble material was removed by centrifugation at 4 °C, and the resulting supernatants were collected. Protein concentrations were determined using the Pierce BCA Protein Assay Kit (#23225, Thermo Fisher Scientific, Waltham, Massachusetts, USA) with bovine serum albumin as the standard. For immunoblot analysis, 30 µg of protein from each sample was mixed with 4× Laemmli loading buffer, denatured at 95 °C for 5 min, and separated by SDS−PAGE on 4−20% gradient Mini-PROTEAN TGX Stain-Free gels (#4568094, Bio-Rad, Hercules, California, USA). Proteins were transferred onto PVDF membranes using the Trans-Blot Turbo transfer system with Trans-Blot Turbo PVDF packs (#1704156, Bio-Rad, Hercules, California, USA). Membranes were blocked in TBST containing 5% BSA for 1 h at room temperature and incubated overnight at 4 °C with primary antibodies against IQCH (dilution 1:500, #STJ194004, Lot: 4RC229RC22, St John's Laboratory, London, UK) or β-actin (dilution 1:1000, #8457S, Lot: 8, Cell Signaling Technology, Danvers, Massachusetts, USA). After washing, membranes were incubated with HRP-linked anti-rabbit IgG secondary antibody (dilution 1:2000, #7074S, Lot: 34,, Cell Signaling Technology, Danvers,

Massachusetts, USA) at room temperature. Immunoreactive signals were detected using SuperSignal™ West Dura chemiluminescent substrate (#34076, Thermo Fisher Scientific, Waltham, Massachusetts, USA) and captured using an Amersham ImageQuant 800 imaging system (#29105868, Cytiva, Marlborough, Massachusetts, USA).

### Statistical analysis
Normality of datasets, excluding cytokine arrays, was assessed using the Kolmogorov-Smirnov or Shapiro-Wilk test. Data conforming to a normal distribution were analyzed using two-tailed unpaired t-test. Results are presented as mean ± SEM. For the analysis of cytokine array data, multiple unpaired t-tests were employed. The false discovery rate was controlled at 5% using the Benjamini-Hochberg correction for multiple comparisons[69]. Statistical calculations were conducted using GraphPad Prism 10 (GraphPad Software, La Jolla, CA; RRID: SCR_002798).

## Data availability
All data generated or analysed during this study are available on Zenodo[88] (https://doi.org/10.5281/zenodo.18163056), except for the raw STARNET which was made accessible by members of the STARNET project. Source data are provided with this paper.

## Code availability
SNEEP for predicting TF-SNVs is available under (https://github.com/SchulzLab/SNEEP/). STARE for regulatory interactions is provided under (https://github.com/SchulzLab/STARE). The software for the colocalization analysis was taken from (https://cran.r-project.org/web/packages/coloc/index.html). Custom scripts used in this manuscript can be found on GitHub (https://github.com/SchulzLab/CADLinc) and are published on Zenodo[89] (https://doi.org/10.5281/zenodo.18241673).

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

## Acknowledgements

The work was funded by the Sonderforschungsbereich SFB TRR 267 (DFG, 403584255, project B05 ZC & HS and project Z03 MHS & DH), the German Research Foundation (DFG 510049865) to ZC, Corona-Stiftung Junior Research Group Grant to ZC, the Deutsche Herzstiftung call for 'Diagnose und Therapie der koronaren Herzkrankheit (KHK) / Herzinfarkt' to ZC and the Federal Ministry of Education and Research (Bundesministerium für Bildung und Forschung, BMBF) as part of the German Center for Child and Adolescent Health (DZKJ) under the funding code 01GL2407A to MW and DT. We acknowledge the support of the German Center for Cardiovascular Research (DZHK) with a Postdoc Start-up Grant (ID: 81X3600510) to ZC, the DZHK (IDs: 81X2600534, 81Z0600501 and 81Z0600506) to ZC and HS, International Cardiovascular Research Partnerships Awards (ICRPA) on project ID-SCAD (Z.C.), the DZHK (IDs: 81Z0200101) to MHS and (81X2200151 to NB), the Cardio-Pulmonary Institute (CPI) [EXC 2026] ID: 390649896 to MHS, the DFG SFB1531 (S03, project ID 456687919) to MHS & NB, the Clinical and Translational Science Awards (CTSA) grant UL1TR004419 from the National Center for Advancing Translational Sciences to LM, and the European Union's Horizon Europe (European Innovation Council) program under grant agreement No 101115381 MIRACLE to HS. This work was partly supported through the computational and data resources and staff expertise provided by Scientific Computing and Data at the Icahn School of Medicine at Mount Sinai. We thank the International Human Epigenome Consortium (IHEC) for providing access to reprocessed and harmonized epigenomic data from a broad collection of human cell and tissue types.

## Author contributions

Study design and supervision by M.H.S and Z.C. Collection of epigenome data, prediction of enhancer-gene interactions and related analyses by D.H. Running SNEEP and analyses on TF-level were done by N.B. GATES test by N.B. and N.K. Co-expression analysis by N.B. and F.B.A. Sequence similarity analysis by R.K.M. Co-localization analysis with eQTL data by L.L. and A.D. Collection of known CAD disease genes by Z.C. Integration of data and analyses on gene-level by D.H. PheWAS analysis and processing of STARNET expression data by A.D.

Collaboration for STARNET data with J.L.M.B. and L.Ma. Biological experiments by X.S. and S.L., for which SGBS preadiocytes were provided by D.T. and M.W. Support for FACS and cytokine experiments by A.M under supervision of H.Sager. Analysis of atherosclerosis plaques data by Z.L. under supervision of L.Maegdefessel. H.Schunkert provided resources related to the CAD summary genetics statistics and suggestions and comments to the project and manuscript. Manuscript writing by D.H., X.S., N.B., A.D., M.H.S., Z.C. Feedback on manuscript from all authors.

## Funding

## Competing interests

The authors declare no competing interests.

## Additional information

https://doi.org/10.1038/s41467-026-70216-6).

