## [Transparent Peer Review file · Nature Communications]

Cell type-specific epigenetic regulatory circuitry of coronary artery disease loci

Corresponding Author: Professor Marcel Schulz

Version 0:

Reviewer comments:

Reviewer #1

(Remarks to the Author)

Can the authors quantify this statement: "CREs with CAD-SNVs shared across cell types were more likely to have a TF-SNV than those unique to a cell type."

Regarding this statement: "39.80% were predicted to affect transcription factor binding (N = 5,397) (Fig. 1D)". Does this mean the TF motif has been compromised or that the SNV overlapped with a predicted TF binding region? I understand from your workflow that these are predicted affects on predicted TF sites, not validated cell type specific TF binding sites from ChIP-seq data. Please clarify these points.

Similarly for this statement: "Vascular and immune cells were the leading cell types having relatively large fractions of CRE- and TF-SNVs, such as endothelial cells (ECs), smooth muscle cells (SMC), fibroblasts, T cells, monocytes, and neutrophils (Fig. 1E)". These are predicted TF binding sites from cell type specific ATAC/DNase data, correct?

Regarding: "Our analysis also pointed to an ncRNA-related pathway, 'miRNA targets in ECM and membrane receptors', for CAD (Fig. 2B), which might represent a novel mechanism for the disease." Did you test whether this pathway is enriched in previously reported CAD SNVs and culprit genes, which would indicate whether it is a newly identified mechanism?

"The binding of 26 TFs was affected in more than one cell type." This should read 'predicted' binding, correct?

Are the observations in Figures 4A and 4B not a result of the way you identified the candidate genes? In other words, the genes were identified based on association with these biomarkers/measurements? If one looks at all gene expression, is the pattern of enrichment across these categories not observed, suggesting enrichment?

One assumes IQCH-AS1 is located near the mRNA gene for IQCH on the chromosome. What is the effect of deleting IQCH-AS1 on IQCH transcript and protein abundance in adipocytes?

Is the expression of IQCH (the mRNA, not the ncRNA) altered in STARNET human cohorts?

(Remarks on code availability)

Reviewer #2

(Remarks to the Author)

Cell type-specific epigenetic regulatory circuitry of coronary artery disease loci

The authors have approached coronary artery disease from an epigenetic angle and characterised over 1500 genes linked to CAD associated SNPs at 1% FDR.

From a reductionist view, one wishes to pin a gene to a variant but in this manuscript, ~900 FDR CAD loci have 1500 genes working together. Not all CAD SNPs end up overlapping with a CRE so the overall implication is pretty overwhelming. It is an analysis in the welcome direction but I feel it could be made clearer and hopefully the suggestions help.

From 72K CAD SNPs, the authors were able to annotate 13.5K which mapped in a CRE. It will be good if the authors would try to go back and allocate genes to the loci so one could start to understand which variants are talking to which genes and how many of the loci can be resolved using this approach. I have had a look and cant find this information in the supplement. Furthermore, this exercise can help prioritize genes that are linked to more strongly associated variants.

CDKN2B-AS1

Supp Fig 2 shows the CREs that are linked to the CDKN2B-AS1 transcript. Can you please show all the FDR SNPs in the locus and other CREs that map to other genes (specially CDKN2A & 2B) to rule out that these genes are not linked to any TF-SNV. ST15 again is difficult to understand.

IQCH-AS1

Fig 4F does not seem to show colocalization. The strongest peak in adipose eQTL does not overlap with WHR GWAS. Please report the PP.H3 & PP.H4. Suppl Table 14 is uninterpretable. Having said that, this CAD locus is a genome wide associated with most likely gene to be SMAD3. It is not clear how the authors came to rs3784699 from fig 4F. The CAD association for this SNP in Aragam et al is $p=0.005$ (not under 1% FDR) so somehow it seems a weak CAD SNP happens to be in LD with a SNP which is a WHR locus and has an eQTL colocalization with IQCH-AS1. You could do a colocalization between SMAD3 & WHR and also WHR & CAD to convince you are looking at the same signal. But at the moment, it seems to be an obesity locus.

PheWAS and colocalization

I understand the authors motivation to perform PheWAS of the identified genes but then performing co-localization seems like a self-fulfilling prophecy. As they are eQTLs (from a particular tissue) they are very likely to co-localize. I think the authors can safely omit colocalization and just report phewas findings.

In general, the manuscript is difficult to follow.

The supplementary material has a lot of information and would benefit from some explanation in the legend of tables & figures and/or by trimming down excessive detail. Eg: Not sure how to get to 8864 CREs or 4233 TF-SNVs from ST4. ST5 in itself contains 6 excel sheets! ST10 can be merged into 1 table instead of 39 separate sheets based on a column which can then be filtered on!

(Remarks on code availability)

Reviewer #3

(Remarks to the Author)

The authors presented a comprehensive integrative analysis of the genetic and epigenetic mechanisms underlying coronary artery disease (CAD). By combining genome-wide association data from over one million individuals with epigenomic profiles from 45 disease-relevant human cell types, they uncovered the cellular regulatory mechanisms by which CAD-associated single-nucleotide variants (SNVs) influence disease risk. This study provides a valuable framework for interpreting non-coding genetic risk via cell-type-specific epigenomic landscapes, offering insights into transcriptional regulation and highlighting non-coding RNAs as key contributors to CAD pathogenesis.

While the findings are compelling, I have several concerns that should be addressed:

1. Statistical Threshold

The authors use CAD signals derived from the 2022 GWAS summary statistics; however, the significance threshold applied appears more lenient than the conventional genome-wide threshold ($P < 5 \times 10^{-8}$). This deviation should be clearly justified, including its rationale and potential implications.

2. Causality and Fine-Mapping

Given the linkage disequilibrium (LD) structure across the genome, the top GWAS signals are not necessarily causal. To address this, fine-mapping and functional annotation should be incorporated to enhance causal inference.

3. Non-Coding RNA Validation

Among 1,580 candidate genes, only one non-coding RNA (IQCH-AS1) has been validated experimentally in vitro. Can the authors expand in vivo investigations of this candidate? For the remaining genes, is high-throughput experimental validation (e.g., CRISPR screens, massively parallel reporter assays) feasible?

4. Ancestry Considerations

Although the GWAS summary statistics were derived from individuals of mixed ancestry, LD pruning and downstream analyses were primarily based on European reference panels. This discrepancy should be discussed in more detail, including how it may impact fine-mapping resolution and generalizability.

5. Tissue Selection

The selection criteria for the 45 disease-relevant tissues should be clarified. Were they chosen based on clinical relevance,

enrichment signals from genomic or epigenomic analyses, or simply based on data availability? A brief justification in the main text would improve transparency.

(Remarks on code availability)

Since the authors were using already publicly available tools in combination, I believe there should be no particular issues.

Version 1:

Reviewer comments:

Reviewer #1

(Remarks to the Author)

The authors have addressed my concerns. My recommendation is that the reviewer figures 1-3 be included in the manuscript supplement.

(Remarks on code availability)

Reviewer #2

(Remarks to the Author)

Thank you for addressing my comments. I am satisfied with the additions and clarifications.

(Remarks on code availability)

Thank you for providing the code.

Reviewer #3

(Remarks to the Author)

The authors have addressed issues raised by my comments, offering solutions and justifications.

While the authors declined the requested in vivo and CRISPR validation due to time constraints and scope, they substituted this with a computational validation, showing 81% of candidates are differentially expressed in clinical datasets (STARNET, Munich Vascular Biobank). This alternative evidence seems robust.

(Remarks on code availability)

Reviewer #1 (Remarks to the Author):

Can the authors quantify this statement: “CREs with CAD-SNVs shared across cell types were more likely to have a TF-SNV than those unique to a cell type.”

We specified this statement and conducted a Fisher's exact test to show that CREs that overlap a CAD-SNV are more likely to be shared across cell types than unique to a cell type. Similarly, we did a test for the TF-SNVs:

“CREs that are shared across cell types were more likely to contain a CAD-SNV or a TF-SNV than CREs unique to a cell type (two-sided Fisher's exact test, both p -values ≤ 0.0001 ; log2-oddsratio for CAD-SNVs 1.46; log2-oddsratio for TF-SNVs 1.68).”

In this context, we restructured the respective paragraph in the manuscript to make it easier to follow.

Regarding this statement: “39.80% were predicted to affect transcription factor binding (N = 5,397) (Fig. 1D)”. Does this mean the TF motif has been compromised or that the SNV overlapped with a predicted TF binding region? I understand from your workflow that these are predicted affects on predicted TF sites, not validated cell type specific TF binding sites from ChIP-seq data. Please clarify these points. Similarly for this statement: “Vascular and immune cells were the leading cell types having relatively large fractions of CRE- and TF-SNVs, such as endothelial cells (ECs), smooth muscle cells (SMC), fibroblasts, T cells, monocytes, and neutrophils (Fig. 1E)”. These are predicted TF binding sites from cell type specific ATAC/DNase data, correct?

We added a more precise description of TF-SNVs:

“Among the CAD-SNVs in CREs (N = 13,563), 39.80% were affecting predicted transcription factor (TF) binding (N = 5,397), meaning that they significantly increase or decrease how well a TF binding motif matches to the DNA sequence (Fig. 1d).” While restructuring this paragraph, we also rephrased the second statement, to make it clearer that we talk about the fraction of CREs in a cell type that contain a TF-SNV: “Vascular and immune cells were the leading cell types with regard to the fraction of their CREs that contain a TF-SNVs, such as endothelial cells (ECs), smooth muscle cells (SMC), fibroblasts, T cells, monocytes, and neutrophils (Fig. 1e), in line with their crucial roles in cardiovascular health. “

Regarding: “Our analysis also pointed to an ncRNA-related pathway, ‘miRNA targets in ECM and membrane receptors’, for CAD (Fig. 2B), which might represent a novel mechanism for the disease.” Did you test whether this pathway is enriched in previously reported CAD SNVs and culprit genes, which would indicate whether it is a newly identified mechanism?

We went back to the pathways enriched in the known CAD genes, and while this miRNA pathway is not enriched among them, five of the known genes are annotated for this term (COL4A1, COL4A2, COL6A3, FN1, LAMB2). We observe enrichment in our novel candidate CAD genes, as we identify two additional genes from this pathway that were not previously identified (TNXB, LAMC1). As a consequence, we removed the statement about it being a potential novel mechanism. Although it has been newly enriched, the genes are not entirely novel.

“The binding of 26 TFs was affected in more than one cell type.” This should read ‘predicted’ binding, correct?

This is correct, thank you for catching that. It now reads ‘The predicted binding of 26 TFs’.

Are the observations in Figures 4A and 4B not a result of the way you identified the candidate genes? In other words, the genes were identified based on association with these biomarkers/measurements? If one looks at all gene expression, is the pattern of enrichment across these categories not observed, suggesting enrichment?

The candidate genes in our study were identified through epigenetic integration, without relying on expression data or eQTL data. Manuscript Figure 4 presents the data of a phenome-wide association study (PheWAS) to validate the disease relevance of the 1580 candidate genes. The PheWAS analysis was performed using tissue eQTLs for the candidate genes to link them to CAD-relevant traits (Supplementary Data 13). Manuscript Figure 4a and 4b show the summary number of genes with tissue eQTLs linked to the tested traits, suggesting strong disease relevance of our candidate genes. Thus, our eQTL-based PheWAS analysis contributes two types of information. Firstly, the tissue eQTL test determines whether a SNP or SNV regulates gene expression (eQTL genes). Secondly, the PheWAS tests if the eQTL genes are associated with CAD-related phenotypic traits.

To answer the last part of your question, we investigated the tissue distribution of the 1,580 genes based on their expression levels in the tissue types used in Manuscript Figure 4. Genes with TPM ≥ 1 in at least 10% of samples for a given tissue were considered. The general expression patterns did not mirror the PheWAS results. For instance, the kidney cortex ranked among the top tissues by the number of expressed genes but showed the lowest PheWAS signal (Revision Figure R1). Moreover, the number of genes expressed in each tissue is substantially higher than that of significant PheWAS associations (Manuscript Figure 4a, b), indicating that our PheWAS findings reflect tissue-specific regulatory effects rather than general expression patterns.

Revision Figure R1: Comparison of tissue enrichment patterns across CAD candidate genes. (A) Distribution of protein-coding and non-coding CAD candidate genes expressed across tissue types. (B) Number of protein-coding and non-coding CAD candidate genes with significant tissue eQTLs associated with the tested GWAS traits.

One assumes IQCH-AS1 is located near the mRNA gene for IQCH on the chromosome. What is the effect of deleting IQCH-AS1 on IQCH transcript and protein abundance in adipocytes?

We thank the reviewer for this important point. We considered the potential effect of CRISPR editing of IQCH-AS1 on IQCH expression and therefore targeted a region that does not overlap any IQCH exons. To further answer your question, we measured IQCH mRNA and protein levels in IQCH-AS1-knockout (KO) adipocytes, respectively by qPCR and Western blotting. We found no changes in IQCH mRNA or protein levels in IQCH-AS1-KO adipocytes, compared to control cells (CTR) (Revision Figure R2). Thus, IQCH-AS1 deletion does not affect IQCH mRNA or protein abundance.

Revision Figure R2: Deletion of *IQCH-AS1* does not change *IQCH* mRNA or protein levels. (A) qPCR analysis shows comparable *IQCH* mRNA levels between *IQCH-AS1* knockout (KO) and control (CTR) adipocytes. (B) Western blot and quantification show unchanged *IQCH* protein levels in KO and CTR cells. Data are mean \pm SEM; ns: not significant.

Is the expression of IQCH (the mRNA, not the ncRNA) altered in STARNET human cohorts?

In the STARNET cohorts, mRNA expression of *IQCH* is significantly lower in subcutaneous (SF) and visceral abdominal (VAF) fat from CAD cases than in control samples (Revision Figure R3A). The trend is similar to that of *IQCH-AS1*. However, the expression level of the lncRNA, *IQCH-AS1*, is higher than that of the host gene, *IQCH* (Revision Figure R3B / Manuscript Figure 5g).

Revision Figure R3: Reduced *IQCH* and *IQCH-AS1* expression in subcutaneous and visceral abdominal fat of atherosclerosis patients. (A) Expression of *IQCH* was reduced in subcutaneous fat (SF) and visceral abdominal fat (VAF) of atherosclerosis patients (n=568 in SF, n=531 in VAF) in comparison to control individuals (n=92 in SF, n=103 in VAF). (B) Same as (A) but for *IQCH-AS1*, taken from Manuscript Figure 5g to ease the comparison. The center line is the median, dashed lines are the upper and lower quartiles.

Reviewer #2 (Remarks to the Author):

From a reductionist view, one wishes to pin a gene to a variant but in this manuscript, ~900 FDR CAD loci have 1500 genes working together. Not all CAD SNPs end up overlapping with a CRE so the overall implication is pretty overwhelming. It is an analysis in the welcome direction but I feel it could be made clearer and hopefully the suggestions help.

From 72K CAD SNPs, the authors were able to annotate 13.5K which mapped in a CRE. It will be good if the authors would try to go back and allocate genes to the loci so one could start to understand which variants are talking to which genes and how many of the loci can be resolved using this approach. I have had a look and cant find this information in the supplement. Furthermore, this exercise can help prioritize genes that are linked to more strongly associated variants.

We appreciate the time that was taken to examine the supplementary tables. Indeed, the information of which TF-SNVs were mapped to which genes was not easily accessible. We appended Supplementary Data 5, which aggregates all information on the gene level, with two additional columns listing the rsIDs and positions of a gene's TF-SNVs. Regarding the genes linked to more strongly associated variants, we appended a column indicating whether a gene would also be identified if only SNVs below the conventional genome-wide significance threshold had been used for the TF-SNV analysis, rather than the FDR-based threshold (see also the first answer to Reviewer 3).

CDKN2B-AS1

Supp Fig 2 shows the CREs that are linked to the CDKN2B-AS1 transcript. Can you please show all the FDR SNPs in the locus and other CREs that map to other genes (specially CDKN2A & 2B) to rule out that these genes are not linked to any TF-SNV. ST15 again is difficult to understand.

By design, our CREs can be linked to multiple genes. Therefore, a CRE overlapping with a TF-SNV may be associated not only with CDKN2B-AS1 but also with additional genes. However, this does not preclude a regulatory effect of the TF-SNV on CDKN2B-AS1. Since we identified CDKN2B-AS1 based on TF-SNVs, we prefer to maintain the current visualization and not show all the FDR-SNVs at the loci. For a more clearly arranged figure, only the lead TF-SNVs were shown in our previous Supplementary Figure 2. In Revision Figure R 4 and the updated Supplementary Figure 3, all 11 TF-SNVs that led to the identification of CDKN2B-AS1 as a candidate CAD gene are shown. CREs linked to CDKN2B-AS1 are marked. Since the reviewer is particularly interested in the genes CDKN2A and CDKN2B, we also highlighted the CREs linked to those genes. Supplementary Data 16 presents all CREs overlapping with a TF-SNV and their target genes separated according to CAD-related cell types. Further, we added a sheet to the table that describes the content of each column.

Revision Figure R4: Detailed overview of *CDKN2B-AS1* loci. The 11 TF-SNVs (upper row, blue dots) with their overlapping CREs (black and red bars) from different cell types are visualized. Red marked CREs are linked to *CDKN2B-AS1*, CREs highlighted with blue boxes are associated with *CDKN2A*, and CREs in orange are linked to *CDKN2B*.

IQCH-AS1

Fig 4F does not seem to show colocalization. The strongest peak in adipose eQTL does not overlap with WHR GWAS. Please report the PP.H3 & PP.H4. Suppl Table 14 is uninterpretable. Having said that, this CAD locus is a genome wide associated with most likely gene to be SMAD3. It is not clear how the authors came to rs3784699 from fig 4F. The CAD association for this SNP in Aragam et al is $p=0.005$ (not under 1% FDR) so somehow it seems a weak CAD SNP happens to be in LD with a SNP which is a WHR locus and has an eQTL colocalization with IQCH-AS1. You could do a colocalization between SMAD3 & WHR and also WHR & CAD to convince you are looking at the same signal. But at the moment, it seems to be an obesity locus.

We thank the reviewer for this important and thoughtful comment. We acknowledge that IQCH-AS1 does not harbor genome-wide significant SNPs for CAD itself, and that the strongest CAD association in this region corresponds to SMAD3, a well-established CAD gene. However, our analysis integrated CAD genetics and epigenetic datasets to identify novel CAD candidate genes among the genes or loci without genome-wide significance in the work of Aragam et al. IQCH-AS1 represents one of the novel CAD candidate genes.

In response to the reviewer's comment, we carefully revisited our analysis and found the weak colocalization of IQCH-AS1 due to the mixed eQTLs from both STARNET and GTEx. We now reperformed GWAS colocalization with tissue eQTLs separately for the GTEx and STARNET datasets to minimize heterogeneity. When multiple tissues showed colocalization of the gene, we chose the one with the largest number of significantly colocalized SNPs for the visualization in Manuscript Figure 4f. We also have refined our analytical pipeline to make the colocalization criteria more stringent by increasing the posterior probability of Hypothesis 4 (PPH4) threshold from 0.6 to 0.8. We observed a strong colocalization signal of IQCH-AS1 eQTLs from visceral abdominal fat tissue (VAF) with body mass index (BMI) (PPH4=0.89) (Manuscript Figure 4f). Importantly, the SMAD3 and IQCH-AS1 loci are located in close proximity but have independent eQTLs contributing to obesity phenotypes in different adipose tissue types — subcutaneous fat (SF) for SMAD3 (Revision Figure R5 A) and visceral adipose fat for IQCH-AS1 (Revision Figure R5 B). These eQTLs colocalize with distinct WHR GWAS peaks (Revision Figure R5 C and R5 D, respectively), indicating tissue-specific regulatory effects at each locus. Additionally, while the WHR GWAS signal is colocalized with both SMAD3 and IQCH-AS1 eQTLs, the BMI GWAS signal is only with IQCH-AS1 VAF eQTLs but not with SMAD3 SF eQTLs (Revision Figure R5 E and R5 F). Overall, our results indicate that the CAD-known gene SMAD3 and novel IQCH-AS1 likely act on obesity and CAD risk via independent mechanisms within the same genomic region.

We also updated the corresponding PheWAS results in Supplementary Data 12 and included all posterior probability values (PPH0–PPH4) for transparency and interpretability in the Supplementary Data 11. Further, Supplementary Table 14, we apologize for the confusion and provide an updated version of the Supplementary Table 14 in the revised version of our manuscript (now Supplementary Data 15).

Revision Figure R5: *SMAD3* and *IQCH-AS1* eQTL-GWAS colocalization. (A) Subcutaneous fat (SF) eQTLs regulating *SMAD3* expression. (B) Visceral abdominal fat (VAF) eQTLs regulating *IQCH-AS1* expression. (C-D) Genetic variants associated with waist–hip ratio (WHR) GWAS, highlighted the lead and LD variants corresponding to eQTL SNPs of *SMAD3* (C) and *IQCH-AS1* (D). (E-F) Genetic variants associated with body mass index (BMI) GWAS, highlighted the lead and LD variants corresponding to eQTL SNPs of *SMAD3* (E) and *IQCH-AS1* (F).

PheWAS and colocalization

I understand the authors motivation to perform PheWAS of the identified genes but then performing co-localization seems like a self-fulfilling prophecy. As they are eQTLs (from a particular tissue) they are very likely to co-localize. I think the authors can safely omit colocalization and just report phewas findings.

Generally, PheWAS evaluates the associations between genetic variants (e.g., SNPs or SNVs) and a wide range of human traits and diseases. Genetic variants within the gene region are taken for this analysis. However, these variants are not necessarily relevant to the host gene. For our PheWAS analysis, we focused on genetic variants regulating gene expression, namely the expression quantitative trait loci (eQTLs), and considered eQTLs of CAD-relevant tissues. This preliminarily excludes the significant PheWAS traits associated with part of the non-causal variants. After our PheWAS analysis, we further applied colocalization analysis to the eQTLs with significant traits, testing the association of a group of LD variants/eQTLs with the traits. This analysis further eliminates potential false positive results. We included an overview of the GWAS–eQTL colocalization workflow in Supplementary Figure 5a / Revision Figure R6, which outlines the key steps from eQTL selection to PheWAS and colocalization.

1. Obtain Ensembl IDs for 1,580 CAD candidates

2. Collect eQTLs for each gene:

- GTEx v8 (13 tissue types)
- STARNET (7 tissue types)

- – eQTLs that regulate Gene expression in Tissue A

3. Integrate eQTLs with GWAS summary statistics:

- extract corresponding SNPs from 29 GWAS datasets
- keep only genome-wide significant associations (p-value < 5e-8)

- – eQTLs from tissue A that are also associated with GWAS trait B

4. Perform colocalization analysis for each tissue-trait pair using the eQTL and GWAS statistics of SNPs from step 3

5. Select significant tissue-trait combinations with PPH4 ≥ 0.8 for each gene

Gene 1
Tissue A – Trait B
Tissue A – Trait C
Tissue D – Trait C

Gene 2
Tissue D – Trait B
Tissue F – Trait C

Gene 3
Tissue A – Trait G
Tissue F – Trait B

Revision Figure R6: Overview of the eQTL-based Phenome-wide association study (PheWAS). In the manuscript shown as Supplementary Figure 5a.

Interestingly, after performing PheWAS, 1,264 of the 1,580 CAD candidate genes showed significant associations. Following colocalization analysis ($PPH4 \geq 0.8$), 1,173 genes remained, indicating that colocalization does not substantially reduce the number of significant genes, but it greatly reduces the number of gene–tissue–trait combinations, from 60,442 before colocalization to 30,265 after. This demonstrates that colocalization helps to refine and prioritize the biological context rather than confirm an expected eQTL-GWAS signal overlap. It allows us to pinpoint the specific tissues where genetic regulation of expression is most likely to be relevant for the trait, improving the interpretability of PheWAS findings.

In general, the manuscript is difficult to follow.

As other reviewers' comments also pointed out that certain aspects were not well explained, we went through the entire manuscript to improve phrasing and clarity. Some paragraphs were reordered to improve readability and others were condensed. In addition, we added Supplementary Figure 1a / Revision Figure R7 and Supplementary Figure 5a (see Revision Figure R5 above) to provide a visual aid for the analysis steps.

Revision Figure R7: Overview of the two complementary approaches used to determine our candidate CAD gene set. In the manuscript shown as Supplementary Figure 1a.

The supplementary material has a lot of information and would benefit from some explanation in the legend of tables & figures and/or by trimming down excessive detail. Eg: Not sure how to get to 8864 CREs or 4233 TF-SNVs from ST4. ST5 in itself contains 6 excel sheets! ST10 can be merged into 1 table instead of 39 separate sheets based on a column which can then be filtered on!

We thank the reviewer for taking the time to go through the supplementary material. We agree that the reader would benefit from a more detailed explanation and added detailed column explanations to the sheets whenever needed (e.g. Supplementary Data 3, 4 and 16). Supplementary Data 4 is the output generated by the SNEEP pipeline. Each row indicates not only whether a SNV is a TF-SNV but also the information which TF is affected, whether it is a predicted increase or decrease of binding, the exact position etc. If more than one TFs binding is affected by a SNV, the SNV appears multiple times in the file. Information regarding the CREs is not included there. We apologize for the confusion and link the table later in the text for clarity.

Supplementary Data 5 indeed contains many different sheets. However, besides the sheet that collects all information for our candidate genes, we believe that the additional sheets are necessary as well. We incorporate multiple gene sets from various sources, and it would not be possible to reproduce our candidate genes without those.

As suggested, we merged Supplementary Data 10 into one table with an additional column holding the TF information.

Reviewer #3 (Remarks to the Author):

The authors presented a comprehensive integrative analysis of the genetic and epigenetic mechanisms underlying coronary artery disease (CAD). By combining genome-wide association data from over one million individuals with epigenomic profiles from 45 disease-relevant human cell types, they uncovered the cellular regulatory mechanisms by which CAD-associated single-nucleotide variants (SNVs) influence disease risk. This study provides a valuable framework for interpreting non-coding genetic risk via cell-type-specific epigenomic landscapes, offering insights into transcriptional regulation and highlighting non-coding RNAs as key contributors to CAD pathogenesis.

While the findings are compelling, I have several concerns that should be addressed:

1. Statistical Threshold

The authors use CAD signals derived from the 2022 GWAS summary statistics; however, the significance threshold applied appears more lenient than the conventional genome-wide threshold ($P < 5 \times 10^{-8}$). This deviation should be clearly justified, including its rationale and potential implications.

Indeed, $P < 5e-8$ is a stricter threshold for significant GWAS SNVs. However, both thresholds of a false discovery rate (FDR) $< 1\%$ and $P < 5e-8$ are commonly used for genetic studies. What often happens is that novel loci from a newer GWAS study, due to an increased sample size, originate from previous FDR GWAS loci based on a smaller sample size. Thus, to

maximize our candidate gene discovery, especially novel genes, we selected SNVs with FDR < 1% from the latest summary genetic statistics of CAD generated from 1,165,720 individuals for our study [1]. We have access to an unpublished meta-analysis of CAD GWAS with a larger sample size of 1,250,780 from the Broad Institute in which we checked the p-values of SNVs that are above genome-wide significance but below the 1% FDR threshold in our summary genetic statistics. 13.7% (2,986 out of 21,842) SNVs become significant (p -value $\leq 5e-08$) in the larger GWAS.

To prioritize CAD candidate genes, we employed two key methods: GATES and SNEEP. The former method is not affected by threshold selection, as it considers all SNVs independent of p-value or FDR. The latter includes a background model to adjust random SNVs (Supplementary Figure 1e). From this aspect, the selection of the threshold is unlikely to affect the reliability of the candidate genes. Nonetheless, we agree that it is interesting to know which genes are regulated by a CAD SNV with $P < 5e-8$. We added this information via an additional column in Supplementary Data 5.

In addition, to assess whether the genes identified exclusively by the more lenient FDR-based threshold are meaningful, we compared the genes that reach the FDR and P-value thresholds in the PheWAS analysis. We found that the genes exclusively identified with the more lenient FDR cutoff are similarly associated with CAD-relevant traits as those found via P-value-based threshold (p -value = 0.21, two-sided Wilcoxon signed-rank test across traits).

Therefore, we think the FDR-based threshold in our analysis is reasonable.

2. Causality and Fine-Mapping

Given the linkage disequilibrium (LD) structure across the genome, the top GWAS signals are not necessarily causal. To address this, fine-mapping and functional annotation should be incorporated to enhance causal inference.

We would like to point out that we use two ways to prioritize genes: the gene-based association test GATES and the tool SNEEP which finds SNVs that affect predicted TF-binding in regulatory regions. The latter thus follows a clear functional hypothesis with respect to CAD gene regulation (workflow visualized in the new Supplementary Figure 1a / Revision Figure R8).

Revision Figure R8: Overview of the two complementary approaches used to determine our candidate CAD gene set. In the manuscript shown as Supplementary Figure 1a.

GATES considers all SNVs in the gene body without any cutoff on the p-value and uses the LD structure of the SNVs to calculate a p-value for a gene. Therefore, fine-mapping is not applicable for GATES.

For SNEEP, we consider all SNVs with 1% FDR (as pointed out above). The SNVs in LD with those 1% FDR SNVs are explicitly added, as the causal relationships are not known. We then use a background model of random SNVs and their proxy SNVs in LD to estimate how often we can expect a gene to have TF-SNVs in its CREs (see Methods section “Identification of TF-SNVs and CAD-TFs”). From this background model, we derive our filtering criteria matching a 1% FDR: a gene needs to have at least two CREs containing TF-SNVs that are not in LD with each other (Supplementary Figure 1e). We believe that our prioritization is quite strict, given multiple steps of thresholding, inclusion of cell type-specific data, and the comparison to a background model that is fitted to the data. Still, we followed the advice and tested a fine-mapping tool, namely the widely used R package SusieR [2]. SusieR (a refined implementation of the Sum of Single Effects model) aims to find credible sets of variants, which are supposed to contain causal SNVs while being as small as possible. Since SusieR starts with a set of SNVs in a locus, we separated the 1% FDR SNVs and their proxies into loci (gap size of $\geq 10\text{kb}$ between loci) and ran their function `susie_rss` to get the posterior inclusion probability (PIP) per SNV (https://stephenslab.github.io/susieR/reference/susie_rss.html). To then map the SNVs with $\geq 95\%$ PIP (4,609 out of the 72,432 SNVs) to eQTL genes from GTEx. More precisely, we selected the eQTL-gene pairs from GTEx that were fine-mapped with CAVIAR [3] (as provided on the GTEx portal) for 13 CAD-relevant tissues and overlapped them with the SusieR SNVs (without a statistical test on the colocalization) [4], [5], [6]. This resulted in only 204 genes, 121 of which are among the candidate CAD genes we present in the manuscript (Revision Figure 9A). A GO functional enrichment showed that the genes found via CAVIAR had fewer terms related to lipid metabolism and endothelial-related terms. For terms of the human phenotype database, our CAD candidate genes were enriched for multiple terms related to arteries and atherosclerosis (‘Abnormal coronary artery morphology’, ‘Coronary artery atherosclerosis’ etc.), while none of those terms were enriched among the CAVIAR genes (Revision Figure 9B). Given the comparatively small number of identified genes and the results of the functional enrichment, we concluded that this combination of fine-mapping with SusieR and eQTL data does not give us more meaningful genes. Appending the two statistical approaches is likely too limiting, even with such a powerful GWAS. Usage of eQTL for linking SNVs to genes also introduces a bias towards specific variants and positions close to transcription start sites [7]. If we were to use eQTLs at this point, the PheWAS that follows would be circular, given that it also uses eQTL data.

Fundamentally, fine-mapping approaches assume that one locus contains one or a handful of causal SNVs without considering biological context [8]. In fact, a locus might have a different function in different cell types and therefore distinct causal SNVs in each cell type [9]. Thus, multiple causal SNVs and genes exist for one locus, which likely contribute to the 1580 CAD candidate genes from a large number of cell types ($N = 45$) in our study.

Revision Figure R9: Candidate genes via fine-mapping with SusieR. (A) Overlap of the candidate CAD genes presented in the manuscript and genes found via SusieR followed by eQTL-gene pairs mapped with CAVIAR from GTEx. (B) Human Phenotype terms enriched among the genes found via SusieR & CAVIAR and the manuscript candidates. Shown are only terms containing any of the following strings: athero, arter, vessel.

With regard to functional annotation of the SNVs, we present an annotation with Ensembl's Variant Effect Predictor in Manuscript Figure 1b. In addition, our annotation of TF-SNVs can also be seen as a functional annotation and we are unsure which other types of functional annotations should be added.

3. Non-Coding RNA Validation

Among 1,580 candidate genes, only one non-coding RNA (IQCH-AS1) has been validated experimentally in vitro. Can the authors expand in vivo investigations of this candidate? For the remaining genes, is high-throughput experimental validation (e.g., CRISPR screens, massively parallel reporter assays) feasible?

We thank the reviewer for this suggestion. However, we believe characterizing additional candidates is out of the scope of this manuscript, given the complexity and extent of the computational aspects of this work. Aside from that, to conduct in vivo experiments in Germany, we must first obtain approval from the relevant government agency. This usually takes 6-8 months. We have confirmed that no mouse model is available for Gm16759 (ENSMUSG00000086539.9), the mouse homolog of IQCH-AS1. The amount of work required to generate this mouse model and perform the in vivo study will necessitate a full PhD training period. We estimate a similarly large workload for CRISPR screening and massively parallel reporter assays. Both experiments require gene library design and commercial synthesis, transgene delivery into cells, deep sequencing at the single-cell level, and sophisticated downstream bioinformatic analysis. We hope the reviewer can understand that, for these reasons, we are unable to perform additional experiments.

However, we agree that additional support from external data sources is important to strengthen the validity of our 1,580 candidate genes. Thus, we investigated how many genes are significantly differentially expressed in CAD cases compared to controls. The following paragraph was added to the manuscript:

“We investigated how many of our candidate genes are significantly differentially expressed in CAD cases compared to controls. To do so, we used two datasets. First, RNA sequencing of atherosclerosis plaques from early (controls) and advanced (cases) patients undergoing carotid endarterectomy (Munich Vascular Biobank [10]). Second, RNA data from five tissues from CAD cases and controls (STARNET [11]), namely visceral abdominal fat, subcutaneous fat, atherosclerotic aortic root, liver, and skeletal muscle (≥ 500 cases and ≥ 100 controls for each tissue). Remarkably, a total of 1,276 of our candidate genes (81%), including 165 non-coding RNA genes, were differentially expressed in at least one tissue (Revision Figure R10 / Manuscript Figure 2d). Many genes were differentially expressed across multiple tissues, with 16% of our candidates being differential in all tissues from both cohorts and 29% in all tissues from STARNET (Supplementary Figure 1d).”

Revision Figure R10: Differential expression of candidate genes in CAD cases and controls. (A) Number of candidate genes that are differentially expressed (DEG, $FDR \leq 5\%$ and absolute $\log_2FC \geq 0.3$) across tissues from STARNET and atherosclerotic plaque from the Munich Vascular Biobank (MVB). Shown as Figure 2d in the manuscript. (B) UpSet plot showing the intersection of candidate genes that are differentially expressed across tissues from STARNET and atherosclerotic plaque from the Munich Vascular Biobank (MVB). Limited to the ten largest intersections. Shown as Supplementary Figure 1d in the manuscript.

In this context, we updated the Figure 3 panels that also use STARNET data to be consistent in how DEGs are defined. Previously, we used an FDR threshold of $\leq 1\%$, which is stringent compared to the more commonly used 5% threshold.

4. Ancestry Considerations

Although the GWAS summary statistics were derived from individuals of mixed ancestry, LD pruning and downstream analyses were primarily based on European reference panels. This discrepancy should be discussed in more detail, including how it may impact fine-mapping resolution and generalizability.

We agree that the population difference between GWAS and the LD data should be discussed, and we added the following statement to the Discussion. We also now explicitly state in the Methods that we used the European ancestry files from the Alkes group for LD information.

“Another point to consider is that the GWAS and LD data are not from the same population. While both are of majorly European ancestry, we might over- or underestimate the linkage of SNVs from other ancestries. Efforts can be invested to explore ancestry-specific epigenetic regulatory circuitry of CAD loci after a large growth of GWAS sample size from other populations, such as African, Hispanic, and Asian.”

5. Tissue Selection

The selection criteria for the 45 disease-relevant tissues should be clarified. Were they chosen based on clinical relevance, enrichment signals from genomic or epigenomic analyses, or simply based on data availability? A brief justification in the main text would improve transparency.

We began the selection by listing cell types that were clinically relevant, and then collected those for which we were able to obtain data. To support their role, we have now added columns to Supplementary Data 1 that list references for their potential involvement in CAD. We added a respective statement in the Results:

“The cell types were manually prioritized based on potential clinical relevance and the final selection determined by data availability.”

References

- [1] K. G. Aragam *et al.*, “Discovery and systematic characterization of risk variants and genes for coronary artery disease in over a million participants,” *Nat. Genet.*, vol. 54, no. 12, pp. 1803–1815, Dec. 2022, doi: 10.1038/s41588-022-01233-6.
- [2] Y. Zou, P. Carbonetto, G. Wang, and M. Stephens, “Fine-mapping from summary data with the ‘Sum of Single Effects’ model,” *PLOS Genet.*, vol. 18, no. 7, p. e1010299, Jul. 2022, doi: 10.1371/journal.pgen.1010299.
- [3] F. Hormozdiari, E. Kostem, E. Y. Kang, B. Pasaniuc, and E. Eskin, “Identifying Causal Variants at Loci with Multiple Signals of Association,” *Genetics*, vol. 198, no. 2, pp. 497–508, Oct. 2014, doi: 10.1534/genetics.114.167908.
- [4] S. K. Han *et al.*, “Mapping genomic regulation of kidney disease and traits through high-resolution and interpretable eQTLs,” *Nat. Commun.*, vol. 14, no. 1, p. 2229, Apr. 2023, doi: 10.1038/s41467-023-37691-7.
- [5] C. Chen *et al.*, “PancanQTLv2.0: a comprehensive resource for expression quantitative trait loci across human cancers,” *Nucleic Acids Res.*, vol. 52, no. D1, pp. D1400–D1406, Jan. 2024, doi: 10.1093/nar/gkad916.
- [6] K. Watanabe, E. Taskesen, A. Van Bochoven, and D. Posthuma, “Functional mapping and annotation of genetic associations with FUMA,” *Nat. Commun.*, vol. 8, no. 1, p. 1826, Nov. 2017, doi: 10.1038/s41467-017-01261-5.

- [7] H. Mostafavi, J. P. Spence, S. Naqvi, and J. K. Pritchard, "Systematic differences in discovery of genetic effects on gene expression and complex traits," *Nat. Genet.*, vol. 55, no. 11, pp. 1866–1875, Nov. 2023, doi: 10.1038/s41588-023-01529-1.
- [8] D. J. Schaid, W. Chen, and N. B. Larson, "From genome-wide associations to candidate causal variants by statistical fine-mapping," *Nat. Rev. Genet.*, vol. 19, no. 8, pp. 491–504, Aug. 2018, doi: 10.1038/s41576-018-0016-z.
- [9] E. Long, J. Williams, H. Zhang, and J. Choi, "An evolving understanding of multiple causal variants underlying genetic association signals," *Am. J. Hum. Genet.*, vol. 112, no. 4, pp. 741–750, Apr. 2025, doi: 10.1016/j.ajhg.2025.01.018.
- [10] J. Pelisek *et al.*, "Biobanking: Objectives, Requirements, and Future Challenges—Experiences from the Munich Vascular Biobank," *J. Clin. Med.*, vol. 8, no. 2, p. 251, Feb. 2019, doi: 10.3390/jcm8020251.
- [11] O. Franzén *et al.*, "Cardiometabolic risk loci share downstream cis- and trans-gene regulation across tissues and diseases," *Science*, vol. 353, no. 6301, pp. 827–830, Aug. 2016, doi: 10.1126/science.aad6970.